# Predictive Differential Training Guided by Training Dynamics

## Abstract

This paper centers around a novel concept proposed recently by researchers from the control community where the training process of a deep neural network can be considered a nonlinear dynamical system acting upon the high-dimensional weight space. Koopman operator theory, a data-driven dynamical system analysis framework, can then be deployed to discover the otherwise non-intuitive training dynamics. Taking advantage of the predictive power of the Koopman operator theory, the time-consuming Stochastic Gradient Descent (SGD) iterations can be bypassed by directly predicting network weights a few epochs later. This novel predictive training framework, however, often suffers from gradient explosion especially for more extensive and complex models. In this paper, we incorporate the idea of differential learning, where different parts of the network can undergo different learning rates during training, into the predictive training framework and propose the so-called "predictive differential training" (PDT) to sustain robust performance for accelerated learning even for complex network structures. The key contribution is the design of an effective masking strategy based on Koopman analysis of training dynamics of each parameter in order to select the subset of parameters that exhibits "good" prediction performance. PDT also includes the design of an acceleration scheduler to keep track of the prediction error so that the training process can roll back to the traditional GD-based approaches to "correct" deviations from off-predictions. We demonstrate that PDT can be seamlessly integrated as a plug-in with existing optimizers, including, for example, SGD, momentum, and Adam. The experimental results have shown consistent performance improvement in terms of faster convergence, lower training/testing loss, and fewer number of epochs to achieve the best loss of Baseline.

## 1 Introduction

The advent of cutting-edge hardware (Li et al., 2014) and the development of parallel processing techniques (Li et al., 2020) have greatly accelerated the training process of the Deep Neural Network (DNN). However, enhancing the fundamental techniques of DNN training continues to be a significant challenge. From the inception of SGD (Robbins & Monro, 1951), which has since become a mainstay in DNN training, numerous techniques have been proposed to increase the efficiency of the underlying optimization task, including, for example, learning rate annealing and momentum (Sutskever et al., 2013), RMSprop (Tieleman & Hinton, 2012), and Adam (Kingma & Ba, 2014). In addition to these first-order optimizers, second-order alternatives (Martens, 2010) utilizing curvature information or second-order derivatives of the loss function have been explored to potentially enable more efficient convergence.

Notably, the Adam optimizer (Kingma & Ba, 2014) has been a significant advancement and belongs to the family of *differential learning*, where different parts of the network can exhibit different learning rates during training. The different parts can be, for example, layer-specific (Devlin et al., 2019; He et al., 2019a) or parameter-specific (Tieleman & Hinton, 2012; Kingma & Ba, 2014; Duchi et al., 2011a). This is particularly useful in large-scale models where different layers or parameters might require different levels of adjustment during training.

Very recently, a novel interpretation of the DNN training process has been proposed, mainly by researchers from the control community (Redman et al., 2022; Dogra & Redman, 2020; Manojlovic

et al., 2020; Tano et al., 2020) – If it is intuitive to consider a pre-trained DNN as an inherently nonlinear static system acting upon the high-dimensional inputs, then *the DNN "training process" itself is a "nonlinear" dynamical system acting upon the high-dimensional "weight space"*! It is a discrete dynamical system since the weights of a DNN evolve over each iteration (or epoch) according to the optimization process adopted. This drastically different interpretation has led to the establishment of a novel mathematical framework for learning. Koopman operator theory (Mezić, 2005), a powerful data-driven dynamical system analysis tool, is often adopted to exploit the underlying dynamics in the seemingly non-intuitive training process of a DNN. Taking advantage of the predictive power of the Koopman operator theory, the time-consuming SGD iterations can be bypassed by directly predicting network weights a few epochs later (Dogra, 2020; Dogra & Redman, 2020; Tano et al., 2020). We refer to these approaches as *predictive training*.

However, practical challenges quickly emerge. The absence of actual gradient descent means that convergence cannot be guaranteed, and the framework is sensitive to disturbances in the weight space, leading to error accumulation across iterations. As the network scales, the Koopman-based prediction training framework becomes increasingly ineffective. This issue is mostly due to the lack of adaptive mechanisms when applying prediction-based acceleration. That is, existing predictive training approaches tend to apply the predicted weights to *all* parameters without considering the different dynamics they might exhibit during the training process. This often leads to gradient explosion, especially for more extensive and complex models.

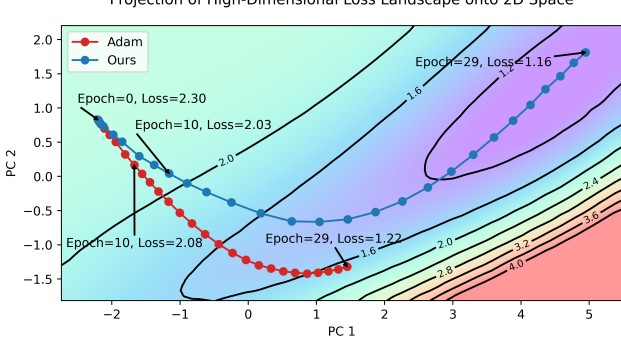

Figure 1: Comparison of training trajectories and loss landscapes between Adam and the proposed PDT. (AlexNet is trained on CIFAR-10)

In this paper, we propose *predictive differential training* (PDT) where acceleration by prediction is applied to only the parameters where we have the high confidence on prediction performance. This selective acceleration is conceptually similar to various adaptive learning rate methods. For instance, Adagrad (Duchi et al., 2011b) targets acceleration at rare features, momentum (Rumelhart et al., 1986) prioritizes weights with the largest recent velocity, and the popular Adam optimizer (Kingma & Ba, 2014) employs a combined strategy. Figure 1 illustrates the compelling effectiveness of PDT over Adam through a visual comparison of the training trajectory and loss landscape. The contribution of the proposed PDT is three-fold:

- We design an effective masking strategy based on Koopman analysis of training dynamics of each parameter and select the subset of parameters that exhibits "good" prediction performance.
- We design a scheduler to keep track of the prediction error so that the training process can roll back to the traditional GD-based approaches to "correct" deviations from off-predictions.
- We demonstrate that PDT can be seamlessly integrated as a plug-in with existing optimizers, including, for example, SGD, momentum, and Adam.

## 2 BACKGROUND AND RELATED WORK

The key notion of Koopman analysis is the representation of a (possibly nonlinear) dynamical system as a linear operator on a typically infinite-dimensional space of functions (Mezić, 2021; 2005; Mezić &

Banaszuk, 2004). Koopman-based approaches directly contrast with standard linearization techniques that consider the dynamics in a close neighborhood of some nominal solution. Indeed, Koopman analysis can yield linear operators that accurately capture fundamentally nonlinear dynamics.

**Koopman Operator Theory.** As a brief description, consider a discrete-time dynamical system $\mathbf{x}_{i+1} = T(\mathbf{x}_i)$, where $\mathbf{x}_i \in \mathbb{R}^n$ is the current state and $\mathbf{x}_{i+1}$ is the next state after application of the potentially nonlinear mapping $T$. Consider also a vector-valued observable $\mathbf{g}(\mathbf{x}) \in \mathbb{R}^m$. The evolution of observables under this mapping can be described according to

$$\mathbf{g}(\mathbf{x}_{i+1}) = \mathbf{g}(T(\mathbf{x}_i)) = \mathcal{K}\mathbf{g}(\mathbf{x}_i). \tag{1}$$

where $\mathcal{K}$ operates on the vector space of observables and maps $\mathbf{g}(\mathbf{x}_i)$ to $\mathbf{g}(\mathbf{x}_{i+1})$. $\mathcal{K}$ is referred to as the "Koopman operator" that is associated with the fully nonlinear dynamical system.

The Koopman operator is linear, following from linearity of the composition operator, but also infinite-dimensional. As such, for dynamical systems with a pure point spectrum for observables (Mezić, 2020), its action can be decomposed according to

$$\mathbf{g}(\mathbf{x}_{i+1}) = \mathcal{K}\mathbf{g}(\mathbf{x}_i) = \sum_{k=1}^{\infty} \lambda_k^{i+1} \phi_k(\mathbf{x}_0)\mathbf{c}_k, \tag{2}$$

where $\lambda_k$ is an eigenvalue associated with the eigenfunction $\phi_k(\mathbf{x})$ evaluated at the initial condition $\phi_k(\mathbf{x}_0)$ and $\mathbf{c}_k$ is the reconstruction coefficient (also referred to as the "Koopman mode") associated with projecting $\mathbf{g}$ onto the eigenspace. It immediately follows that

$$\mathbf{g}(\mathbf{x}_{i+\tau}) = \sum_{k=1}^{\infty} \lambda_k^{\tau} \phi_k(\mathbf{x}_i)\mathbf{c}_k \tag{3}$$

for any $\tau \in \mathbb{N}$. Eq. 3 provides a convenient and general framework to "predict and control" a given dynamical system. Each Koopman mode evolves over time with its frequency and decay rate governed by the imaginary and real components, respectively.

Koopman-based techniques are particularly useful in a data-driven setting because they only require measurements of observables. As such, they can be implemented even when the underlying model dynamics are unknown.

**Dynamic Mode Decomposition (DMD).** When using Koopman-based approaches, it is critical to identify a suitable *finite* basis for representing the infinite-dimensional Koopman operator. Dynamic Mode Decomposition (DMD) (Schmid, 2010) is one standard approach for inferring Koopman-based models. It uses least-squares fitting techniques to approximate a finite-dimensional linear matrix operator, $A$, that advances high-dimensional measurements of a system forward in time:

$$\mathbf{g}(\mathbf{x}_{i+1}) \approx A\mathbf{g}(\mathbf{x}_i) \tag{4}$$

where $A$ is an approximation of the Koopman operator, $\mathcal{K}$ in Eq. 1 restricted to a measurement subspace spanned by direct measurements of the state $\mathbf{x}$. Since the weight space of a neural network is a *fully observable* system, we define $\mathbf{g}(\mathbf{x})$ to be the identity function in this work. That is, $\mathbf{w}_i = \mathbf{g}(\mathbf{x}_i)$. In practice, we often use "snapshots" of the system arranged into two data matrices, $W_i$ and $W_{i+1}$, where columns of these two matrices indicate measurements (i.e., network weights) taken at a certain time, and $W_{i+1}$ is $W_i$ shifted by one time step. Hence,

$$W_{i+1} \approx AW_i, \tag{5}$$

and $A$ can be solved by

$$A = W_{i+1}W_i^{\dagger} = W_{i+1}V\Sigma^{-1}U^T \tag{6}$$

where $W_i = U\Sigma V^T$ is the Singular Value Decomposition (SVD), and $W_i^{\dagger}$ denotes the pseudo-inverse of $W_i$. A comprehensive discussion of DMD and its related variants has been provided in (Kutz et al., 2016).

**DNN Training as a Dynamical System.** There have been a few works in recent years that adopt Koopman-based approaches to accelerate the training process of a general-purpose DNN model (Dogra & Redman, 2020; Tano et al., 2020; Manojlovic et al., 2020). (Dietrich et al., 2020) is generally considered the first work that establishes the connection between Koopman operator theory

and acceleration of numerical computation. (Dogra, 2020) is also one of the pioneer works but with a focus specifically on neural networks for solving differential equations. Generally speaking, these works take advantage of the prediction capability of the Koopman operator theory framework, as shown in Eq. 3, to directly predict network weights a few epochs later, thus bypassing the time-consuming SGD iterations. However, we show in Fig. 2 that these methods tend to fail for larger network structures as the prediction horizon for Koopman-based approaches decreases and as network size increases.

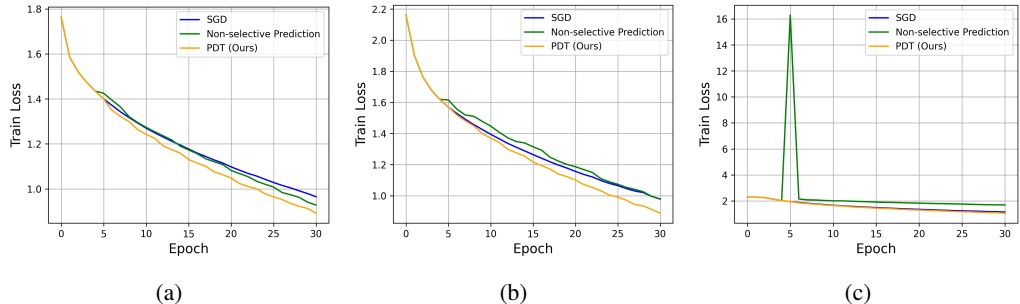

(a)                  (b)                  (c)

Figure 2: Performance comparison on CIFAR-10 using fully connected (FC) networks with varying depths, between SGD, PDT, and Koopman-based predictive training where the predicted weights are applied to all parameters without checking the prediction quality (Tano et al., 2020). Batch size=256, lr=0.01. (a) 2-layer FC network. (b) 4-layer FC network. (c) 6-layer FC network. In our setup, for every three epochs of SGD, predictions are performed for the next five steps. Subsequently, training reverts to SGD to potentially rectify minor errors introduced by the predictions.

The proposed PDT, largely due to its adaptive attention to different training dynamics from different parameters, is able sustain network growth. The efficiency of PDT has been validated on several benchmark models (e.g., AlexNet, ResNet, and ViT) and datasets (e.g., CIFAR-10 and ImageNet).

## 3 METHODS

In this section, we elaborate on the proposed Koopman-based predictive differential training (PDT) framework. We first describe the rationale of the proposed masking strategy that identifies the subset of weights with "good" predictions. This is followed by a discussion of the acceleration schedule.

### 3.1 CONSTRUCTING THE MASK

We can apply Eq. 7 to predict future measurements of $\mathbf{w}$ over $\tau$ epochs.

$$\mathbf{w}_{i+\tau} = A^\tau \mathbf{w}_i \tag{7}$$

where $A$ can be calculated from Eq. 6. The challenge, however, is how to determine if this prediction is "good" or "bad".

In fact, the correlation between quality of prediction and training dynamics has been heavily studied. From neuroscience perspective, the quality of predictions made by neurons is intricately linked to their learning dynamics (Schultz et al., 1997; Friston, 2010). Accurate predictions lead to more stable and efficient learning, while poor predictions drive stronger synaptic adjustments to improve future performance.

We design a masking strategy that is based on the following two principles.

- The quantity criterion: The absolute weight change between the predicted weight and the current weight should be *larger* than the absolute weight change from the one-step optimization (e.g., using SGD) to enable accelerated learning.

- The direction criterion: The direction of weight change from prediction should be consistent with that from optimization. That is, if the optimization procedure would result in a

weight decay or weight increase, then the predicted weight should correspondingly decay or increase.

Based on these two principles, a mask, $\mathbf{m}$ can be constructed with its element equals to 1 if both Eqs. 8 and 9 are satisfied; otherwise the corresponding element is zero,

$$\|\mathbf{w}_{i+\tau}^{\text{pred}} - \mathbf{w}_i^{\text{pred}}\| > \|\mathbf{w}_{i+1}^{\text{opt}} - \mathbf{w}_i^{\text{opt}}\|, \text{ the quantity criterion} \tag{8}$$

$$(\mathbf{w}_{i+k}^{\text{pred}} - \mathbf{w}_{i+k-1}^{\text{pred}}) \cdot (\mathbf{w}_{i+1}^{\text{opt}} - \mathbf{w}_i^{\text{opt}}) > 0, k = \{1, \cdots, \tau\}, \text{ the direction criterion} \tag{9}$$

Note that Eq. 9 is a rigid criterion to enforce not only the final predicted weight changes along the same direction as the one-step optimization outcome, but that each intermediate predicted weight all change in the same direction.

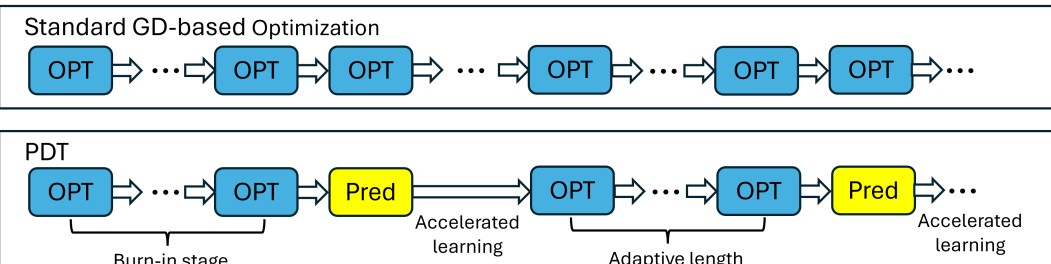

Figure 3: Comparison of the standard SGD-based optimization and the proposed PDT framework in accelerating training.

## 3.2 ACCELERATION SCHEDULE

The acceleration schedule concerns the problem of "when" to enable prediction. As illustrated in Fig. 3, the "prediction" block is strategically placed among the regular SGD optimization blocks, acting as a plug-in enhancement within the existing optimization framework. The placement of the "prediction" block is solely determined by the masking strategy designed in Sec. 3.1. If no element in the mask is qualified as "good" prediction, then standard SGD-based optimization takes place; otherwise, qualified predicted weights will be incorporated to accelerate learning. This approach is analogous to the "one-step-ahead" strategy employed by NAG (Nesterov, 1983), where the subsequent step of standard optimization acts to correct any small errors that may arise from the Koopman prediction.

Let us use a toy example to demonstrate the effect of accelerating the learning of a subset of variables to further motivate the concept of differential learning. Consider the function,

$$f(x, y, z, u, v, w) = x^2 + y^2 + \sin(z) + u^2 - \cos(v) + w^2 + xy + y\sin(z) + uvw,$$

which involves six variables: $x, y, z, u, v, w$. To find the minimum of this function, we employ a simple gradient descent optimization process. Starting from the initial point $[2.0, 2.0, 1.0, 0.5, -0.5, 1.5]$ with a learning rate of $0.01$, it takes 67 steps to converge to a minimum.

We then explore an alternative optimization strategy where the variables $x, y, z$ undergo an optimization process that is three times faster than the standard process, while $u, v, w$ are optimized at the normal rate but employing the updated values of $x, y, z$. See Fig. 10 in Appendix A.1 for the acceleration trajectory, where the trajectory maintains the same direction for $x$ and $y$ but achieves convergence in just 27 steps.

This example shows that by strategically identifying a subset of variables and simply increasing their learning rate, the training can be accelerated by about 60%.

Figure 4 further illustrates how qualified predicted weights and standard SGD-derived weights are mixed together to achieve accelerated learning as showcased in the toy example.

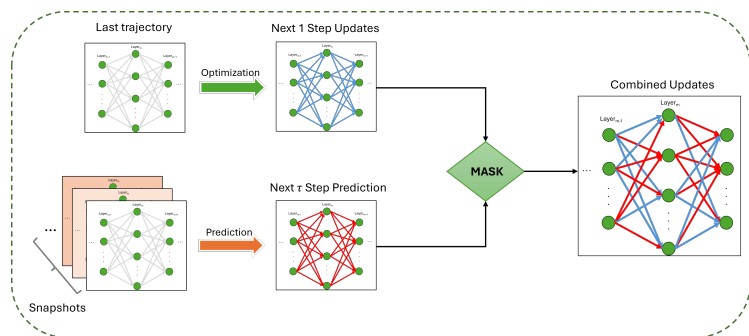

Figure 4: Illustration of one PDT-based optimization step.

### 3.3 COMPUTATIONAL COMPLEXITY ANALYSIS

To facilitate our discussion, we consider a DNN with $N$ parameters. The computational load for processing each batch is directly proportional to both the batch size ($B$) and the number of parameters ($N$), resulting in a complexity of $\mathcal{O}(B \times N)$ per batch. When extended to the entire dataset with $S$ samples across one epoch, the complexity scales to $\mathcal{O}(S \times N)$.

Integrating Koopman operator predictions into the DNN training process entails constructing a data matrix from $h$ past epochs of the parameter trajectories, with the matrix dimensions being $N \times h$. The primary computational burden arises from performing SVD on this matrix with a complexity of $\mathcal{O}(N \times h^2)$. Given that $N$ significantly exceeds $h$ — with $h$ usually being a small number like 5 to 10, and $N$ potentially reaching the millions or even billions—the quadratic impact of $h$ remains manageable relative to $N$.

Since Koopman predictions are integrated at much less frequent intervals than standard batch processing—potentially at epoch-level intervals—this approach can lead to significant computational savings and efficiency enhancements in the training of large-scale neural networks. A detailed analysis of computational efficiency with experimental results is provided in Sec. A.4.

## 4 EXPERIMENTS

We conduct four sets of experiments to evaluate the effectiveness of the proposed PDT framework in accelerating learning. The first set of experiments implements PDT across a variety of popular neural network architectures using a range of popular optimizers and evaluate the savings in run-time. The second set investigates the effectiveness of the proposed masking strategy. The third set evaluates the proposed masking strategy against other potential metrics for prediction quality, like validation loss. The final set of experiments studies the effect of some important hyperparameters.

### 4.1 GENERALIZATION STUDY OF PDT

We implement the proposed Koopman-based PDT process across a variety of popular neural network architectures, including Fully-Convolutional-Network (FCN), AlexNet, ResNet, and ViT-Base. We also use a range of optimizers, including the SGD, SGD with momentum, and Adam.

In all experiments, we use the past five epochs to form the snapshot with a one-epoch interval to predict weights in the next five steps. Prediction is initiated starting from the 5th epoch. As elaborated in Sec. 3, the computational load of the Koopman-related calculations is comparable to that of batch-level updates. However, since we apply these calculations at the epoch level, the overhead introduced by the DMD is effectively compensated by the acceleration in loss reduction. We observe from both Table 1 and Fig. 5 that the proposed PDT consistently achieves the best training loss of the Baseline in fewer number of epochs without sacrificing performance. All experiments were repeated with five random seeds (0, 100, 200, 300, 400) to ensure reliability.

The last column in Fig. 5 illustrates a so-called "masked ratio curve" unique to PDT, where it tracks the percentage of predictions accepted according to the masking strategy described in Sec. 3.1. We

observe that the masked ratio always starts with higher values in the early stage of the training process, then generally decreases as training progresses. More interestingly, we observe that smaller networks on simpler tasks (FCN/AlexNet on CIFAR-10) show a relatively more gradual reduction in the masked ratio, while larger networks on more complex tasks (ResNet-50/ViT on ImageNet) exhibit a much sharper reduction of masked ratio, especially at the early stage of the training process. This pattern implies that for larger networks on larger datasets, the training dynamics is more complex and challenging to predict at the initial training stage, resulting in a rapid reduction of the percentage of weights that can be convincingly predicted (according to the proposed masking strategy). The training process of a deep network with millions to billions of parameters indeed presents an intriguing dynamical system that the control community has not faced before. This would stimulate further investigation into the development of better data-driven dynamical system analysis algorithms in addition to DMD.

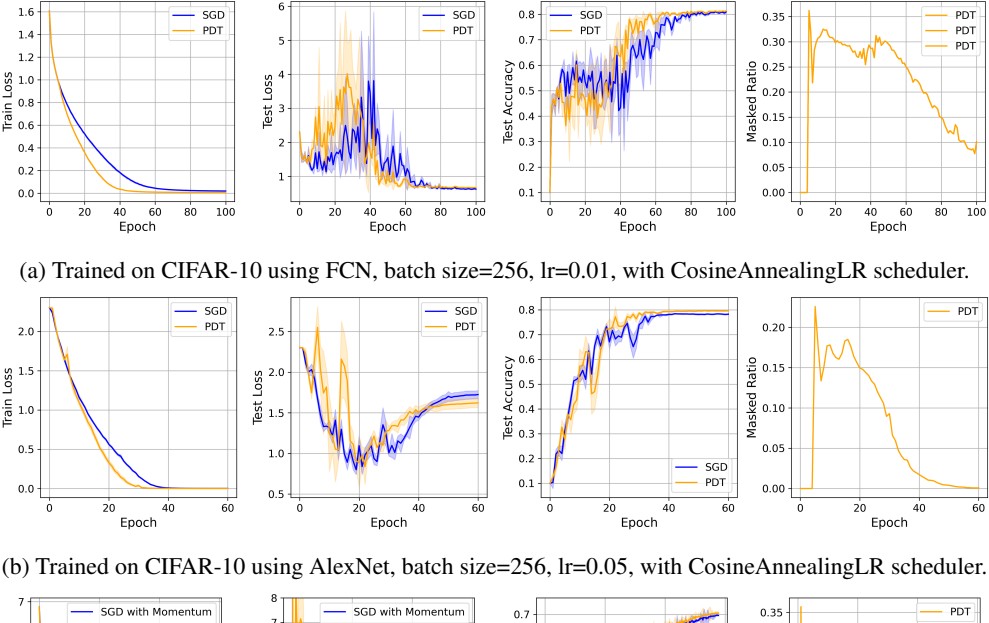

(a) Trained on CIFAR-10 using FCN, batch size=256, lr=0.01, with CosineAnnealingLR scheduler.

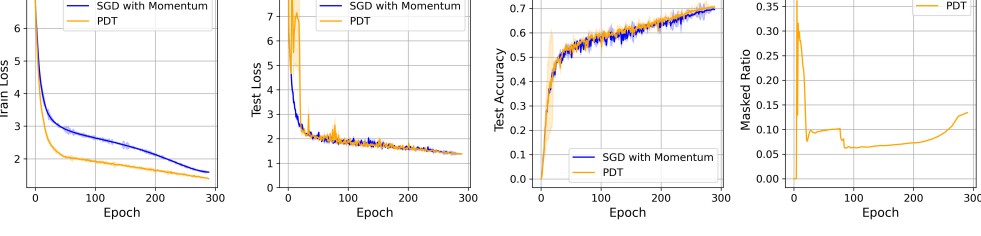

(b) Trained on CIFAR-10 using AlexNet, batch size=256, lr=0.05, with CosineAnnealingLR scheduler.

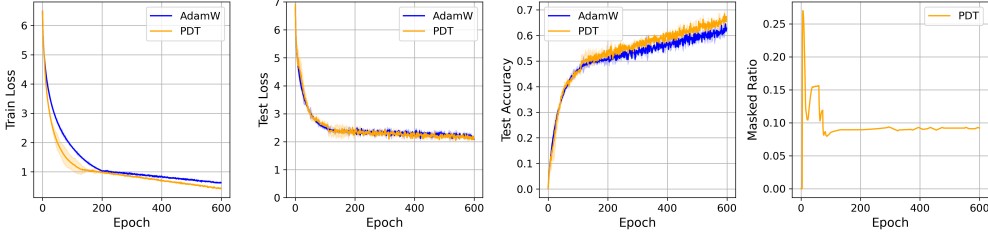

(c) Trained on ImageNet-1K using ResNet-50, batch size=600, lr=0.1, momentum=0.9, with CosineAnnealingLR scheduler.

(d) Trained on ImageNet-1K using ViT-Base, batch size=600, lr=0.003, momentum=0.9, with CosineAnnealingLR scheduler.

Figure 5: Performance comparison between baseline optimization and PDT. Note that all the experiments are repeated with 5 different random seeds.

Table 1: Runtime comparison. FCN and AlexNet are trained on a single Nvidia RTX A6000 GPU, while ResNet-50 and ViT-Base are trained on three Nvidia H100 (80 GB) GPUs. Using the same experimental setup and hyperparameter configurations as in Fig. 5.

| Model | Time to Baseline Best Loss (s) | | Runtime per Epoch (s) | | Runtime Reduction (%) |
|---|---|---|---|---|---|
| | Baseline | PDT | Baseline | PDT | |
| FCN | 2145.36 | 1294.74 | 21.45 | 27.86 | 39.65 |
| AlexNet | 675.04 | 424.39 | 11.17 | 12.14 | 37.13 |
| ResNet-50 | 110063.72 | 88752.33 | 379.53 | 422.63 | 19.36 |
| ViT-Base | 259241.21 | 232810.62 | 432.79 | 541.42 | 10.20 |

## 4.2 MASKING STRATEGY

In this experiment, we study the effectiveness of the proposed masking strategy by comparing it with two other strategies, 1) randomly selecting a subset of weights and increase its learning rates, and 2) randomly selecting a subset of predicted weights.

**Comparison with Randomly Selected Acceleration Subsetw.** We conduct an experiment to compare PDT against the strategy of randomly selecting subsets of weights and increasing their learning rates. Figure 6 illustrates each trial's outcomes, with regions highlighted in green showing results from different runs where subsets of weights had their learning rates increased to match the step number used in the predictions. The selection ratio used here matches the average masking ratio applied during PDT. The results clearly indicate that randomly accelerating weights cannot match the performance improvements seen with PDT. Moreover, random selection often leads to significant instability during training.

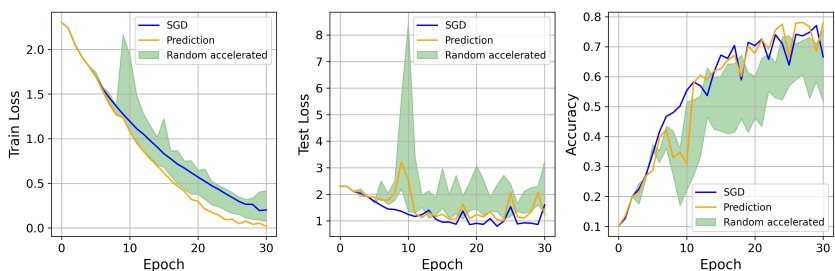

Figure 6: Comparison between PDT and randomly selected subsets with higher learning rates (with the same mask ratio). Trained on CIFAR-10 using AlexNet, batch size=256, lr = 0.05.

**Comparison with Randomly Selecting Predicted Weights.** We perform a series of runs where subsets of Koopman predicted weights are randomly selected and applied to a large network. The regions highlighted in green in Fig. 7 show the outcomes of these trials. Quite frequently, these runs result in gradient explosions, leading to non-recoverable errors (NaN values) in subsequent epochs. This experiment underscores the importance of a thoughtful masking strategy in Koopman Training. Random masking, without considering the training dynamics can lead to severe divergence and training failure. Our findings highlight that strategic selection based on "good" predictions is crucial to the success of PDT.

## 4.3 ACCELERATION SCHEDULE BASE ON VALIDATION LOSS?

Although DMD can make long-term predictions, mismatches with the true evolutionary path of the network weights can occur at any future step, potentially leading to suboptimal training outcomes. According to (Tano et al., 2020), validation loss can be utilized as a criterion to determine optimal points for switching between DMD and SGD during training. Inspired by this strategy, we implement a reference scheduling scheme that switches between prediction and SGD based on the validation loss trend: apply prediction when validation loss decreases and switch back to SGD updates when

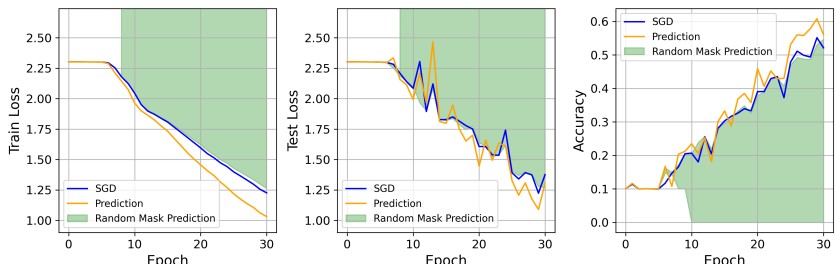

Figure 7: PDT vs. random mask prediction (with the same mask ratio). Trained on CIFAR-10 using AlexNet, batch size=256, lr = 0.01.

validation loss starts to increase. Figure 8 illustrates the training dynamics under this strategy. Initially, DMD is engaged due to its slight advantage in reducing validation loss. However, as training progresses, a significant surge in loss is observed, suggesting a misalignment between the DMD-predicted weights and the optimal trajectory for the network. Even after reverting to SGD, the model failed to recover its performance, indicating that relying solely on validation loss as a trigger for switching between PDT and SGD might be inadequate.

This experiment highlights the complexity of training dynamics and the challenges in using PDT effectively within a traditional training framework. It suggests that while validation loss can serve as an initial indicator for employing advanced predictive methods like DMD, it may not be sufficient on its own to guarantee stable and effective training convergence.

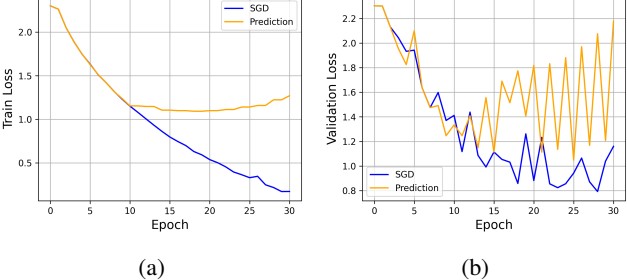

(a)                                          (b)

Figure 8: Performance comparison on CIFAR-10 using AlexNet: SGD vs. Koopman-based prediction (switching between prediction and SGD based on validation loss). (a) Train loss. (b) Validation loss.

## 4.4 EFFECT OF HYPERPARAMETERS

Several primary hyperparameters require careful consideration in our model:

**Prediction Steps ($\tau$):** Derived from DMD, the number of prediction steps significantly influences the training speed. As shown in Fig. 9(a), training accelerates within a certain range of prediction steps. However, extending beyond a critical threshold, such as nine steps in our study, can introduce large errors and potentially cause gradient explosion.

**Prediction Interval (Ti):** The interval between Prediction blocks impacts the effectiveness of acceleration, as depicted in Fig. 9(b). A shorter interval can enhance training speed if the predictions are accurate. Nevertheless, the quality of predictions may decline as the training progresses, rendering the network more sensitive to errors, particularly as it nears convergence.

**Starting Epoch (T0):** The starting epoch for acceleration must be greater than or equal to the number of epochs used to build the snapshot, as illustrated in Figure 9(c). The initiation of acceleration is influenced by factors such as initialization, learning rate, and model architecture.

**Past Snapshot Counts (h):** Figure 9(d) indicates that the number of epochs needed to construct the snapshot matrix for prediction also influences the train loss. This value cannot be too small or

too large. If it is too small, the snapshot will not have sufficient measurements to precisely estimate the dynamics of the training process. On the other hand, if the number of epochs is too large for constructing the snapshot, then DMD would have missed the local dynamics with only a coarser grasp of the general training dynamics.

In addition, a comprehensive study of PDT's performance under different training configurations can be found in Sec. A.3, demonstrating robust performance across various training hyperparameters.

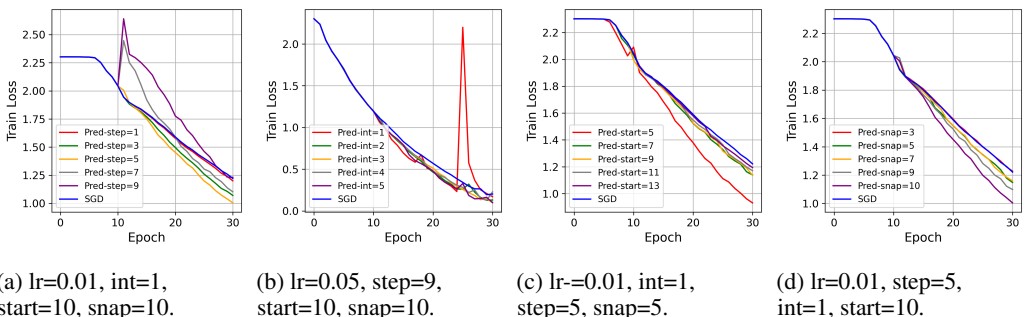

(a) lr=0.01, int=1, start=10, snap=10.
(b) lr=0.05, step=9, start=10, snap=10.
(c) lr-=0.01, int=1, step=5, snap=5.
(d) lr=0.01, step=5, int=1, start=10.

Figure 9: The influence of different parameters. (a) prediction steps, (b) prediction interval, (c) starting epoch, (d) past snapshot counts. Trained on CIFAR-10 using AlexNet, batch size=256.

# 5 DISCUSSION AND CONCLUSION

This paper proposed a novel predictive differential training (PDT) framework based on the study of training dynamics, where we consider the training process as a dynamical system acting upon the weight space. PDT presents stable performance in accelerating training even for complex network structures due to its selective incorporation of predicted weights.

**Future Work and Challenges.** Despite these advancements, considerable work remains. First, further studies into different predictive methods beyond DMD is necessary. Innovative approaches, such as streaming DMD (Hemati et al., 2014; Liew et al., 2022), can not only reduce the memory footprint of constructing trajectory matrices, but also improve computational efficiency.

Second, investigating the impact of PDT on the properties of the learned function, such as loss surface sharpness or smoothness, is highly valuable (Humayun et al., 2024; Foret et al., 2020). These metrics provide a deeper understanding of the model's robustness and generalization capabilities. Based on the current experimental results, we hypothesize that the selective application of predictions may help avoid sharp local minima by allowing more exploration in the weight space. In future work, we intend to incorporate these measures into our analysis to provide a more comprehensive evaluation of PDT, and further explore how these properties influence the efficacy of PDT.

Third, we observe from the masked ratio vs. epoch curves in Fig. 5 that as training prolongs and as training loss converges to a stable value, we should expect the training dynamics to be less complex or easier to predict, which should have resulted in a higher masked ratio. However, in reality, except for the ResNet-50 on ImageNet-1K curve where a small bouncing back on the masked ratio is observed toward the end of the training process, all the rest scenarios exhibit a stable masked ratio, much lower as compared to that at the beginning of the training process. In addition, we would have expected the epoch number, where the masked ratio starts turning flat, to be consistent with that when the training loss enters a plateau, but this is only observed in the complex network scenarios Fig. 5(c) and (d), but not the simple network cases Fig. 5(a) and (b). This seems to indicate that the masked ratio curves can have the potential of indicating when the network overfits, that when the marked ratio starts to drastically decrease again after the initial reduction. This would serve as a potential indicator for early stopping conditions. Although this is out of the scope of the current paper, the potential impact warrants further investigation.

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

# A APPENDIX

## A.1 CONVERGENCE PATH OF THE TOY EXAMPLE

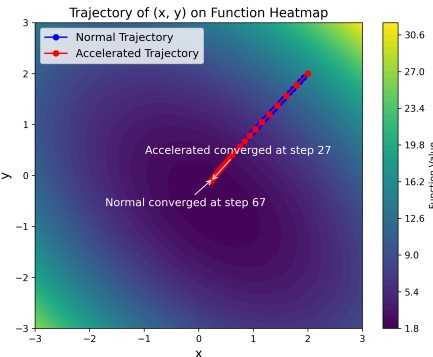

Figure 10: The differential learning trajectory of the toy example provided in Sec. 3.2. Only the $x$ and $y$ dimensions are shown.

## A.2 ALGORITHM DESCRIPTIONS

---

**Algorithm 1** PDT algorithm

---

**Require:** baseline optimizer $O_{base}$, past snapshots counts $h$, start epoch for prediction $T_0$, predicted steps $\tau$, prediction interval $T_i$
**Ensure:** Trained model parameters $\mathbf{w}$
1: Initialize weight history matrix $\mathbf{W}_{N \times h}$, counter $c_e = 0$
2: **for** epoch $t = 0$ to $T$ **do**
3:    **if** $t \geq T_0$ **and** $c_e \geq T_i$ **then**
4:       Obtain $\mathbf{w}_{opt}(t-1)$ from $\mathbf{W}_{N \times h}$
5:       Train model for one epoch using $O_{base}$, save weights after training as $\mathbf{w}_{opt}(t)$
6:       Calculate DMD from $\mathbf{W}_{N \times h}$
7:       Predict future weights from $\mathbf{w}_{pred}(t)$ to $\mathbf{w}_{pred}(t + \tau - 1)$
8:       Create mask $M$ based on $\mathbf{w}_{opt}(t-1)$, $\mathbf{w}_{opt}(t)$, $\mathbf{w}_{pred}(t)$ ... $\mathbf{w}_{pred}(t + \tau - 1)$ (Eq. 8 and 9)
9:       Assemble new weights $\mathbf{w}(t)$ using mask $M$ to combine $\mathbf{w}_{opt}(t)$ and $\mathbf{w}_{pred}(t)$
10:      Update model parameters with updated $\mathbf{w}(t)$
11:      $c_e \leftarrow 0$
12:    **else**
13:       Train model $M$ normally for one epoch using $O_{base}$
14:       $c_e \leftarrow c_e + 1$
15:    **end if**
16:    Update weight history matrix $\mathbf{W}_{N \times h}$
17: **end for**

---

## A.3 EFFECT OF TRAINING HYPERPARAMETERS

To thoroughly evaluate the effectiveness and robustness of PDT under different training configurations, we conduct comprehensive experiments across different learning rates from 0.001 to 0.1 (0.001, 0.01, 0.05, 0.1) and batch sizes from 32 to 512 (32, 64, 128, 256, 512). All experiments were repeated with five random seeds (0, 100, 200, 300, 400) to ensure statistical significance. All experiments are performed on AlexNet with the CIFAR-10 dataset, using SGD as the baseline optimizer and trained for 60 epochs. The PDT-related hyperparameters mentioned in Sec. 4.4 were set to prediction step=5, prediction interval=1, start epoch=5, and past snapshot counts=5.

Table 2: Impact of learning rates and batch sizes on PDT performance. Trained on CIFAR-10 using AlexNet. Note: bold numbers indicate the best performance and underlined numbers indicate the second best performance for each column.

| Batch Size | lr | Method | Final Accuracy (mean ± std) | Best Train Loss (mean ± std) | Time to Baseline Best Loss (s) (mean ± std) | Runtime Reduction (%) |
|---|---|---|---|---|---|---|
| 32 | 0.001 | SGD | 0.6981 ± 0.0458 | 0.6376 ± 0.0127 | 1232.29 ± 4.45 | 40.64 |
| | | PDT | 0.6903 ± 0.0885 | 0.2724 ± 0.0166 | 731.52 ± 12.84 | |
| | 0.01 | SGD | 0.8118 ± 0.0041 | 0.0046 ± 0.0008 | 1194.89 ± 21.09 | 24.25 |
| | | PDT | **0.8146 ± 0.0048** | 0.0021 ± 0.0012 | 905.07 ± 120.51 | |
| | 0.05 | SGD | 0.8049 ± 0.0053 | 0.0156 ± 0.0029 | 1180.72 ± 12.31 | **64.57** |
| | | PDT | 0.8020 ± 0.0052 | 0.0149 ± 0.0073 | 418.38 ± 0.00 | |
| | 0.1 | SGD | 0.1000 ± 0.0000 | 0.3346 ± 0.0098 | 1172.49 ± 39.08 | - |
| | | PDT | 0.1000 ± 0.0000 | 0.3364 ± 0.0132 | - | |
| 64 | 0.001 | SGD | 0.5384 ± 0.0173 | 1.2295 ± 0.0261 | 902.16 ± 19.68 | 35.82 |
| | | PDT | 0.5329 ± 0.1152 | 0.8798 ± 0.0257 | 578.99 ± 55.74 | |
| | 0.01 | SGD | 0.7850 ± 0.0226 | 0.0087 ± 0.0030 | 800.35 ± 5.39 | 23.32 |
| | | PDT | 0.8140 ± 0.0021 | 0.0015 ± 0.0010 | 613.70 ± 8.80 | |
| | 0.05 | SGD | 0.8067 ± 0.0035 | 0.0051 ± 0.0016 | 798.20 ± 3.50 | 27.54 |
| | | PDT | 0.8029 ± 0.0029 | 0.0045 ± 0.0006 | 578.36 ± 16.48 | |
| | 0.1 | SGD | 0.6442 ± 0.2733 | 0.0484 ± 0.0522 | 910.37 ± 18.03 | 56.23 |
| | | PDT | 0.7976 ± 0.0033 | 0.0218 ± 0.0011 | **398.48 ± 21.34** | |
| 128 | 0.001 | SGD | 0.2882 ± 0.0212 | 1.8456 ± 0.0300 | 812.42 ± 21.20 | 17.48 |
| | | PDT | 0.2951 ± 0.0440 | 1.6972 ± 0.0272 | 670.37 ± 23.93 | |
| | 0.01 | SGD | 0.7825 ± 0.0065 | 0.0675 ± 0.0052 | 661.09 ± 6.35 | 14.68 |
| | | PDT | 0.8009 ± 0.0062 | 0.0058 ± 0.0008 | 564.02 ± 16.35 | |
| | 0.05 | SGD | 0.7969 ± 0.0093 | 0.0039 ± 0.0017 | 662.48 ± 7.73 | 9.15 |
| | | PDT | 0.8011 ± 0.0067 | 0.0016 ± 0.0017 | 601.86 ± 17.78 | |
| | 0.1 | SGD | 0.7916 ± 0.0027 | 0.0083 ± 0.0014 | 803.93 ± 3.07 | 8.20 |
| | | PDT | 0.7863 ± 0.0087 | 0.0096 ± 0.0016 | 737.97 ± 0.00 | |
| 256 | 0.001 | SGD | 0.1171 ± 0.0092 | 2.2991 ± 0.0011 | 747.83 ± 20.30 | 7.08 |
| | | PDT | 0.1453 ± 0.0213 | 2.2979 ± 0.0026 | 694.91 ± 14.63 | |
| | 0.01 | SGD | 0.6989 ± 0.0301 | 0.5814 ± 0.0147 | 660.37 ± 0.71 | 19.98 |
| | | PDT | 0.7450 ± 0.0236 | 0.1855 ± 0.0172 | 528.41 ± 7.26 | |
| | 0.05 | SGD | 0.7931 ± 0.0034 | **0.0004 ± 0.0003** | 648.39 ± 8.57 | 21.71 |
| | | PDT | 0.7916 ± 0.0016 | 0.0015 ± 0.0014 | 507.62 ± 11.36 | |
| | 0.1 | SGD | 0.3742 ± 0.3359 | 0.0508 ± 0.0576 | 771.77 ± 3.06 | - |
| | | PDT | 0.3796 ± 0.3425 | 0.0012 ± 0.0011 | - | |
| 512 | 0.001 | SGD | 0.1170 ± 0.0251 | 2.3017 ± 0.0005 | 748.44 ± 42.46 | 6.23 |
| | | PDT | 0.1377 ± 0.0288 | 2.3020 ± 0.0001 | 701.82 ± 23.31 | |
| | 0.01 | SGD | 0.5710 ± 0.0203 | 1.1920 ± 0.0238 | 671.28 ± 9.03 | 18.89 |
| | | PDT | 0.5985 ± 0.0078 | 0.8311 ± 0.0252 | 544.46 ± 12.10 | |
| | 0.05 | SGD | 0.7717 ± 0.0038 | 0.0311 ± 0.0174 | 668.59 ± 7.30 | 10.11 |
| | | PDT | 0.7669 ± 0.0237 | 0.0034 ± 0.0014 | 601.01 ± 44.11 | |
| | 0.1 | SGD | 0.3721 ± 0.3332 | 0.0648 ± 0.0735 | 768.97 ± 3.12 | - |
| | | PDT | 0.4420 ± 0.3420 | 0.0373 ± 0.0155 | - | |

The results in Table 2 show the impact of different batch sizes and learning rates on the performance of PDT. At lower learning rates (0.001, 0.01, and 0.05), PDT consistently outperforms SGD in terms of convergence speed across different batch sizes. PDT shows a significant reduction in the runtime to reach baseline best loss, with an average runtime reduction of 22.76% compared to SGD. For higher learning rates (0.1), both SGD and PDT struggled to achieve stable training, and PDT's advantage

over SGD became less pronounced. Sometimes PDT can significantly reduce the convergence time (for example, when batch size = 64), but other times the accuracy will drop significantly after reaching a high point, or even result in gradient explosion. This suggests that the high learning rate introduced significant stochasticity, reducing the effectiveness of PDT's prediction mechanism. Smaller batch sizes (32, 64) generally achieve more significant runtime reductions.

To address the stability issues observed at higher learning rates and larger batch sizes, different from the previous fixed learning rate, we investigated the effectiveness of the learning rate scheduler. We tested the Cosine Annealing learning rate scheduler with a minimum learning rate of 1e-3. Taking batch size 256 as an example, we observe significantly improved stability and performance. The results are shown in Table 3. The results are particularly noteworthy at higher learning rates (lr=0.1), where the previous experiments in Table 2 show considerable variance. With the cosine annealing scheduler, PDT achieves consistent accuracy improvements across all learning rates while maintaining substantial runtime reductions.

Table 3: Impact of learning rates on PDT performance. Trained on CIFAR-10 using AlexNet, batch size=256, with CosineAnnealingLR scheduler, minimum learning rate 1e-3. Note: bold numbers indicate the best performance and underlined numbers indicate the second best performance for each column.

| Batch Size | lr | Method | Final Accuracy (mean ± std) | Best Train Loss (mean ± std) | Time to Baseline Best Loss (s) (mean ± std) | Runtime Reduction (%) |
|---|---|---|---|---|---|---|
| 256 | 0.001 | SGD | 0.1217 ± 0.0126 | 2.2991 ± 0.0011 | 757.66 ± 26.54 | 9.88 |
| | | PDT | 0.1461 ± 0.0213 | 2.2980 ± 0.0025 | 682.79 ± 2.13 | |
| | 0.01 | SGD | 0.6451 ± 0.0102 | 0.9276 ± 0.0212 | 745.97 ± 47.19 | **41.54** |
| | | PDT | 0.6974 ± 0.0073 | 0.5853 ± 0.0159 | 436.07 ± 16.09 | |
| | 0.05 | SGD | 0.7852 ± 0.0016 | 0.0020 ± 0.0001 | 675.04 ± 27.56 | 37.13 |
| | | PDT | 0.7936 ± 0.0030 | 0.0006 ± 0.0001 | **424.39 ± 20.40** | |
| | 0.1 | SGD | 0.7930 ± 0.0023 | **0.0002 ± 0.0000** | 665.27 ± 9.08 | 19.67 |
| | | PDT | **0.7978 ± 0.0032** | **0.0002 ± 0.0000** | 534.41 ± 12.64 | |

To further investigate PDT's compatibility with different optimization methods, we compare its performance when integrated with different optimizers (SGD, SGD with momentum, and Adam) while keeping the network architecture and other configurations fixed. For SGD with momentum, we set the momentum factor to 0.9. All experiments are conducted on AlexNet with CIFAR-10 using batch size 256, maintaining the same PDT hyperparameters as in previous experiments. The learning rate is 0.1 for SGD, 0.001 for SGD with Momentum, 0.0005 for Adam. The results are shown in Table 4.

Table 4: Impact of baseline optimizers (SGD, SGD with Momentum, and Adam) on PDT performance. Trained on CIFAR-10 using AlexNet, batch size=256, momentum=0.9, with CosineAnnealingLR scheduler. Note: bold numbers indicate the best performance and underlined numbers indicate the second best performance for each column.

| lr | Method | Final Accuracy (mean ± std) | Best Train Loss (mean ± std) | Time to Baseline Best Loss (s) (mean ± std) | Runtime Reduction (%) |
|---|---|---|---|---|---|
| 0.1 | SGD | 0.7930 ± 0.0023 | 0.0002 ± 0.0000 | 665.27 ± 9.08 | 19.67 |
| | PDT | 0.7978 ± 0.0032 | 0.0002 ± 0.0000 | 534.41 ± 12.64 | |
| 0.001 | Momentum | 0.6672 ± 0.0068 | 0.8609 ± 0.0166 | 752.74 ± 9.62 | **41.06** |
| | PDT | 0.7298 ± 0.0051 | 0.5358 ± 0.0165 | **443.68 ± 8.75** | |
| 0.0005 | Adam | 0.7952 ± 0.0063 | **0.0001 ± 0.0000** | 779.13 ± 11.81 | 14.87 |
| | PDT | **0.8050 ± 0.0050** | 0.0002 ± 0.0000 | 663.28 ± 15.30 | |

A.4 ANALYSIS OF COMPUTATIONAL EFFICIENCY

To provide a detailed analysis of PDT's computational efficiency, we compare the computational cost in terms of FLOPs (Floating-point operations per second) between the baseline optimizer and PDT.

Fig. 11 shows the training dynamics with respect to both epochs and total computation cost (measured in TFLOPs). The experiments in Fig. 11 are conducted on AlexNet with CIFAR-10 using batch size of 256, learning rate of 0.05, with Cosine annealing scheduler. While the per-epoch computation of PDT is slightly higher (69.71 TFLOPs) than that of SGD (56.74 TFLOPs) due to the additional DMD calculations and prediction operations, it achieves faster convergence in terms of total computation. Specifically, PDT requires 2596.30 TFLOPs to reach the baseline's best loss, compared to SGD's 3404.32 TFLOPs, representing a 23.74% reduction in computational cost. Moreover, PDT achieves better final accuracy (79.70% vs 78.75%) despite using fewer FLOPs to reach convergence.

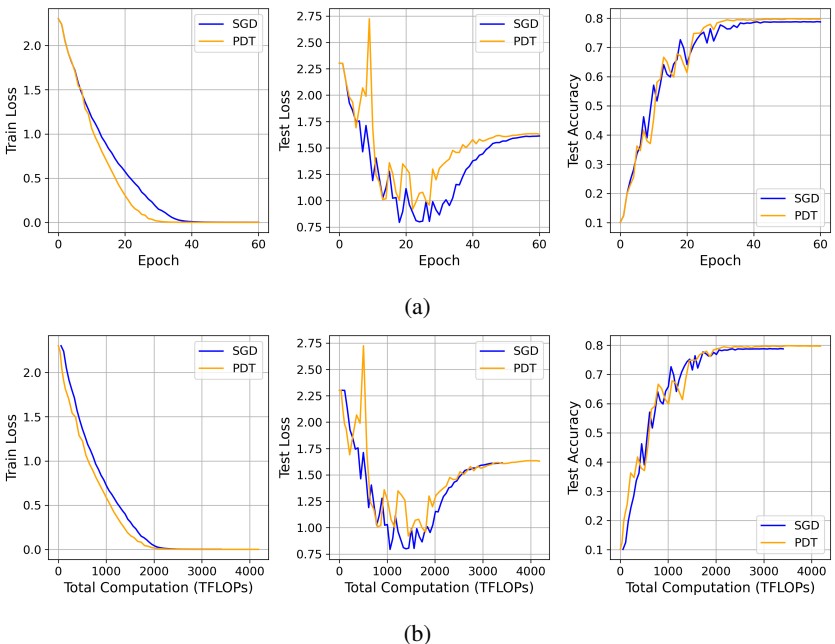

Figure 11: Performance comparison between baseline optimization and PDT, with (a) epochs and (b) TFLOPs as x-axis. Trained on CIFAR-10 using AlexNet, batch size=256, lr=0.05, with Cosine Annealing scheduler.

The results also validate our design choice of keeping the past snapshot count ($h$) small (set to 5 in our experiments). Even with this small $h$ value, which minimizes the computational cost of DMD calculations, PDT achieves substantial acceleration in terms of FLOPs.

### A.5 CONVERGENCE BEHAVIOR USING COSINEANNEALINGLR SCHEDULER IN IMAGENET TRAINING

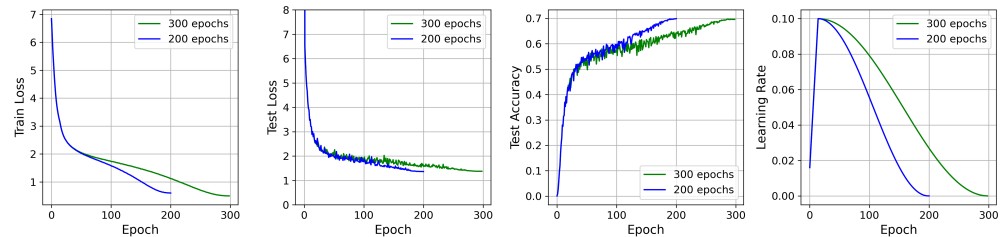

Figure 12: A comparison of training ResNet-50 on ImageNet-1k for 200 and 300 epochs, with the same hyperparameters configuration. Using SGD with Momentum as the optimizer, batch size=600, lr=0.1, momentum=0.9, with CosineAnnelingLR scheduler.

In our experiments, we employed the Cosine Annealing Learning Rate Scheduler, which is designed to gradually reduce the learning rate to near zero as training progresses. This scheduling method

ensures continuous parameter refinement, even in the later stages of training, albeit at a much slower rate (He et al., 2019b). The slow improvement in test loss towards the end of training reflects ongoing refinement rather than a lack of convergence.

To illustrate this behavior, we conducted experiments comparing the training of ResNet-50 on ImageNet-1k for 200 epochs and 300 epochs under identical configurations, SGD with Momentum as the optimizer, batch size=600, lr=0.1, momentum=0.9, with CosineAnnelingLR scheduler. The results in Fig. 12 show that the final performance between these two runs is marginal. This suggests that the model has already achieved sufficient training.

### A.6 EFFECT OF NON-I.I.D. TRAINING DATA

We further investigate the robustness of PDT under some challenging training conditions. For example, when the batch is too small for a diverse dataset like ImageNet, the weight updates could be chaotic since each consecutive batch is no longer an identical distribution. There are two experimental designs that can test this: 1) test PDT on a very large dataset like ImageNet-22K and 2) design a batching scheme to intentionally violate the i.i.d. assumption of mini-batches using a smaller dataset such as CIFAR-10. In the second design, we maintain the normal batch size, but only put samples of the same class in the batch. We also randomize the batch sequence instead of using any fixed order so that there is no regular training set dynamics that DMD might pick up on.

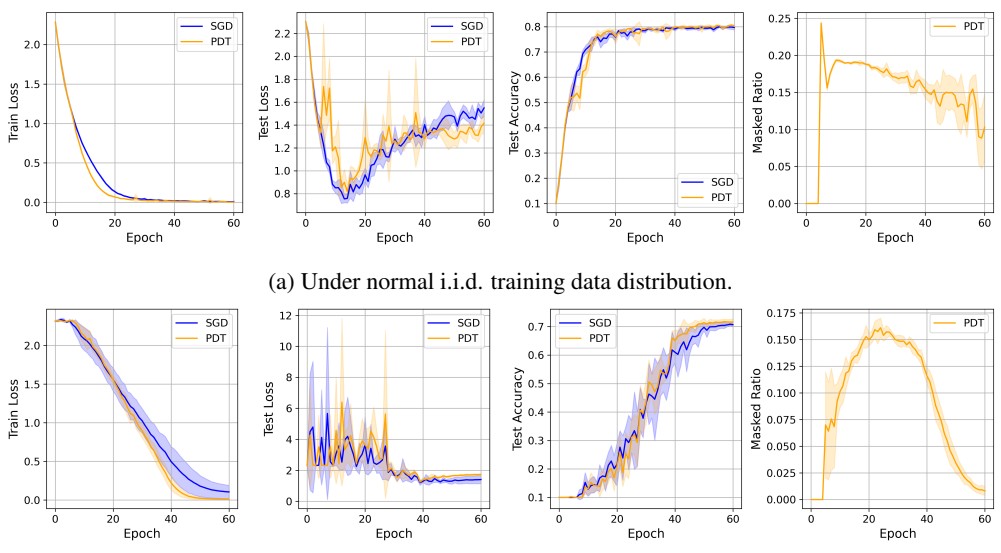

(a) Under normal i.i.d. training data distribution.

(b) Under non-i.i.d. training data distribution.

Figure 13: Performance comparison between SGD and PDT under i.i.d. and non-i.i.d. training data distributions, with the same hyperparameters configuration. Trained on CIFAR-10 using AlexNet, batch size=128, lr=0.05, with CosineAnnelingLR scheduler. The shaded areas represent the standard deviation across 5 runs with different random seeds (0, 100, 200, 300, 400).

Table 5: Performance and runtime comparison between SGD and PDT under i.i.d. and non-i.i.d. training data distributions, with the same hyperparameters configuration. Trained on CIFAR-10 using AlexNet, batch size=128, lr=0.05, with CosineAnnelingLR scheduler.

| Training Data Distribution | Method | Final Accuracy (mean ± std) | Best Train Loss (mean ± std) | Time to Baseline Best Loss (s) (mean ± std) | Runtime Reduction (%) |
|---|---|---|---|---|---|
| i.i.d. | SGD | 0.7969 ± 0.0093 | 0.0039 ± 0.0017 | 662.48 ± 7.73 | 9.15 |
|  | PDT | 0.8011 ± 0.0067 | 0.0016 ± 0.0017 | 601.86 ± 17.78 |  |
| non-i.i.d. | SGD | 0.7067 ± 0.0062 | 0.1053 ± 0.0874 | 806.83 ± 13.15 | 27.90 |
|  | PDT | 0.7159 ± 0.0103 | 0.0119 ± 0.0057 | 581.73 ± 19.34 |  |

Figure 13 and Table 5 show the performance and runtime comparison between SGD and PDT under the non-i.i.d. setting using the second experimental design since non-i.i.d. is guaranteed. We preserve the original i.i.d. sampling of the test set. All experiments are repeated with five random seeds (0, 100, 200, 300, 400) to ensure statistical significance.

We make some interesting observations. First, despite the challenging non-i.i.d. setup, PDT still achieves better performance than SGD in terms of faster convergence without sacrificing accuracy. However, we also observe that in the non-i.i.d. case, learning starts out much more slowly for both SGD and PDT and both take longer to converge. Second, in the non-i.i.d. case, the variance of each of the performance curves is generally larger than those of the i.i.d. case. This is because the model needs to handle more abrupt transitions between different class distributions.

Figure 13 and Table 5 further demonstrate that PDT's advantage extends beyond standard i.i.d. training conditions, showing its robustness to challenging data sets where traditional assumptions about data distribution are violated.

