# OpenReview forum: "Predictive Differential Training Guided by Training Dynamics"
_ICLR.cc/2025/Conference — Submitted to ICLR 2025_

### Official Review · Reviewer_Yfav · 2024-11-04

**Soundness:** 2
**Presentation:** 2
**Contribution:** 2
**Rating:** 5
**Confidence:** 3

**Summary:**

The paper proposes predictive differential training (PDT) of deep neural networks based on Koopman operator theory for accelerated training. The key contribution of the paper involves designing a masking strategy that can stabilize predictive training and enable PDT to reach good network performance. PDT can be integrated within the standard stochastic gradient descent-based (SGD) iterative training where an acceleration scheduler was added to revert to SGD training based on an error in weight prediction. Experiments were presented to investigate the generalizability of PDT, and the effectiveness of the proposed masking strategy.

**Strengths:**

The presentation of the paper is clear. The authors provided the key background needed to understand their method. The authors presented an intuitive understanding of the method which is easy to understand. The authors provided experiments demonstrating the effectiveness of their method and ablation studies to understand the key aspects.

**Weaknesses:**

1.	The main weakness of the paper is the weak empirical performance of PDT. The generalizability study (Fig 5) seems to show similar performance vs SGD or Adam on the test loss and accuracy for the first three rows. The only observable gain seems to be on ImageNet-1k with VIT where it shows better accuracy over AdamW after epoch 200. Probably a table with the final accuracy by running each method should be presented.
2.	The test loss in the first two rows of Fig 5 seems to be indicating overfitting for both approaches hence the effectiveness of the method is unclear.
3.	There are substantial details missing about how the authors have selected the hyperparameters for the baseline optimizers. In my view, the methods should be compared after running an extensive hyperparameter optimization, especially w.r.t the learning rate. The authors also did not provide ablation studies regarding changing learning rate and batch size which is directly correlated with the time taken by SGD or Adam to converge.
4.	In my view, the experimental section needs a lot of updating. I am unclear about the necessity of section 4.3 in the main paper.

**Questions:**

1. In Fig 5, what is the last column is it train or test accuracy?

2. It is unclear if replication studies were performed. If yes, please include the sds.

---

> ### Author Response · Authors · 2024-11-24
> **Response to Reviewer Yfav (Part1)**
>
> Thanks for your valuable review. We will address your concerns one by one in the following.
>
> **Weakness 1: The performance of PDT**
>
> We appreciate the reviewer's careful analysis of our experimental results. In this paper, we focus our studies on the potential of the proposed PDT in accelerating training, i.e., using smaller number of epochs to achieve the best training loss reported by baseline, without sacrificing performance (e.g., test accuracy). Please note that we develop PDT such that it can be plugged in any existing optimization framework, i.e., the baseline. We understand that this may have led to some confusion, especially regarding final accuracy. We added Table 2 and Table 3 to Appendix A.3, to thoroughly evaluate the effectiveness and robustness of PDT under different training configurations. All experiments were repeated with five random seeds, and the mean and standard deviation for all metrics are reported. PDT consistently outperforms the SGD baseline in terms of convergence speed across different batch sizes and learning rates, without sacrificing accuracy of the baseline.
>
> **Weakness 2: The concern about overfitting**
>
> The reviewer was right that both the baseline and PDT in Figs. 5(a) and 5(b) show some overfitting behavior, which is intentional. As stated in response to Weakness 1) above,  our primary focus in these experiments was to compare the convergence speed and training dynamics between PDT and baseline under identical conditions. In these two experiments, we intentionally let the training continue beyond the optimal stopping point to compare if PDT still exhibit the same behavior as that of the baseline. We have conducted additional experiments with various settings in Appendix A.3. The results consistently show that PDT maintains its convergence advantages. This advantage is particularly evident when dealing with larger datasets like ImageNet, as shown by the results on ResNet-50 and ViT-Base, where overfitting is less of a concern.
>
> **Weakness 3: The selection of training hyperparameters**
>
> We have taken the reviewer’s comments seriously and have added a new section Appendix A.3, “Effect of Training Hyperparameters” to thoroughly evaluate the effectiveness and robustness of PDT under different training configurations. We include comprehensive experimental results across different learning rates from 0.001 to 0.1 (0.001, 0.01, 0.05, 0.1) and batch sizes from 32 to 512 (32, 64, 128, 256, 512). The results are shown in Tables 3 and 4 in Sec. A.3. All experiments were repeated with five random seeds (0, 100, 200, 300, 400) to ensure statistical significance.
>
> From Table 2, we see that at lower learning rates (0.001, 0.01, and 0.05), PDT consistently outperforms SGD in terms of convergence speed across different batch sizes. PDT shows a significant reduction in the runtime to reach baseline best loss, with an average runtime reduction of 22.76% compared to SGD. For higher learning rates (0.1), both SGD and PDT struggled to achieve stable training, and PDT's advantage over SGD became less pronounced, suggesting the high learning rate introduced significant stochasticity, reducing the effectiveness of PDT's prediction mechanism. Smaller batch sizes (32, 64) generally achieve more significant runtime reductions.
>
> To address the stability issues observed at higher learning rates and larger batch sizes, we further investigated the effectiveness of the learning rate scheduler and tested the Cosine Annealing learning rate scheduler with a minimum learning rate of 1e-3. The results are shown in Table 3. The results are particularly noteworthy at higher learning rates (lr=0.1), where the previous experiments in Table 2 show considerable variance. With the cosine annealing scheduler, PDT achieves consistent accuracy improvements across all learning rates while maintaining substantial runtime reductions, even with larger batch sizes.
>
> **Weakness 4: The necessity of section 4.3**
>
> The intention behind this section was to explore an alternative acceleration schedule that uses validation loss as an indicator for when to switch between DMD-based predictions and SGD. This approach is adopted in (Tano et al., 2020), which is the state-of-the-art prediction-based acceleration approach that leverages Koopman analysis. In Fig. 2, we already compared how the non-selective prediction approach would fail for larger networks. In this section, we further showed that using validation loss is another reason prediction without selection approaches would fail for larger networks.

---

> > ### Comment · Reviewer_Yfav · 2024-11-24
> >
> > I appreciate the ablation studies performed by the authors and including the hyperparameter details. I would request the authors to highlight in bold the numbers of their tables for ease of reading the numbers.
> >
> > In Fig 5, I understand the authors' logic for showing the overfitted training on CIFAR-10 and indeed it shows PDT following a similar trend as the baseline. However, I still think that calculating the time to best loss for an overfitted training is not useful. There can be pathological reasons (e.g. high learning rate) where the training can show overfitting from a very early stage. For this reason, I believe that the ImageNet training is more relevant in the paper. However, I am still unclear about the authors' comment that on ImageNet the PDT shows any advantage. Visually the test loss in the last two rows seems to show that both SGD and PDT training have not converged after 300 and 600 epochs respectively. Therefore the time to best loss should be the same for both. Please let me know what I am missing. Also, I would request the authors to show the standard dev for the training trajectory.
> >
> > I will increase my score since the authors have included extensive ablation studies.

---

> > > ### Author Response · Authors · 2024-11-27
> > >
> > > Thank you for the prompt feedback and further suggestions. We appreciate the opportunity to address and incorporate these suggestions.
> > >
> > > **1. Bold numbers in tables:**
> > >
> > > Fixed!
> > >
> > > **2. Adding the standard deviation for the training trajectory:**
> > >
> > > We have updated Fig. 5 to include std to the training and testing trajectories.
> > >
> > > **3. Advantages of PDT on ImageNet:**
> > >
> > > The motivation of PDT is NOT to improve accuracy. Rather, it aims to speed up convergence without sacrificing accuracy. That is, if we embed PDT to any baseline, we expect the PDT-based training to converge in less number of epochs than that of the baseline, which is why we use the “Time to baseline best loss” as a metric in Tables 1~4. This metric directly demonstrates PDT's acceleration capability - it shows that PDT can achieve the same level of performance as the baseline's best result, but in less time. Taking ViT-Base with ImageNet-1k as an example [Fig. 5(d)], AdamW (baseline) achieves its best train loss at the end of training (around epoch 600), while PDT reaches this same loss value significantly earlier (around epoch 430). While PDT's per-epoch runtime is slightly higher (541.42s vs. 432.79s), it reaches the baseline's best performance in 10.2% less total runtime, as shown in Table 1. PDT’s ability to reduce the time to reach the best loss of the baseline is confirmed in all of our experiments. Also, see the first column of Fig. 5(a)-(d).
> > >
> > > **4. The overfitted training on CIFAR-10:**
> > >
> > > We agree that experiments on ImageNet may be more meaningful.  We started with AlexNet and CIFAR-10 such that we could carry out extensive experiments to provide some general insights. From Table 2 in Appendix A.3, we can see that when the learning rate is 0.001 and 0.01, no overfitting occurs during the entire training process, and in this case, PDT still outperforms the baseline optimizer. Table 2 also shows that PDT consistently outperforms baseline in terms of convergence speed across different batch sizes, learning rates, and baseline optimizers without sacrificing accuracy.
> > >
> > > **5. The convergence of our ImageNet experiments:**
> > >
> > > The reviewer’s doubt is indeed valid - the test loss in Figs. 5(c) and 5(d) appear to exhibit a downward trend even after 300 epochs and 600 epochs. We would like to clarify that the training curves that don't completely flatten out actually do not always suggest that the model has not converged. In our experiments, we employed the Cosine Annealing learning rate scheduler, which is designed to gradually reduce the learning rate to close to zero (or a very small number) as training progresses. This ensures that the model continues to refine its parameters even in later epochs, albeit at a much slower rate. Such a schedule is particularly effective in avoiding premature convergence to suboptimal solutions and has been shown to improve final test accuracy in prior work. Please see Figure 3 in (He et al., 2019) [<https://arxiv.org/abs/1812.01187>].  The slow improvement in test loss observed towards the end of training reflects ongoing refinement rather than a lack of convergence. In fact, when training ResNet-50 on ImageNet using the cosine annealing scheduler with the same configuration (Using SGD with Momentum as the optimizer, batch size=600, lr=0.1, momentum=0.9), we observed that running for 300 epochs produced a final accuracy almost identical to that achieved after 200 epochs. Therefore, although there may be room for improvement, the models in our experiments have already been sufficiently trained, and the results can reflect the advantages of PDT. We have added Fig.12 in Appendix A.5. to illustrate this behavior.
> > >
> > > > He, T., Zhang, Z., Zhang, H., Zhang, Z., Xie, J., & Li, M. (2019). Bag of tricks for image classification with convolutional neural networks. In Proceedings of the IEEE/CVF Conference on Computer Vision and Pattern Recognition (pp. 558-567).

---

> > > > ### Comment · Reviewer_Yfav · 2024-12-02
> > > > **Appreciate the discussion**
> > > >
> > > > I appreciate the author's detailed response. I now understand the purpose of the work better. However, I am still unconvinced by the time to best loss evaluations of the overfitted CIFAR-10 model. The ImageNet results are promising but they do not convince me because of the decreasing validation loss. I want to thank the authors for the discussion and will leave the final evaluation to the ACs.

---

> ### Author Response · Authors · 2024-11-24
> **Response to Reviewer Yfav (Part2)**
>
> **Question 1: Is the last column in Fig. 5 train or test accuracy?**
>
> The last column (now the 3rd column, as we added the masked ratio to the last column) in Fig 5 shows test accuracy. We have revised Fig. 5 (3rd column) to reflect this.
>
> **Question 2: If replication studies were performed**
>
> To clarify, all experiments were repeated 5 times with different random seeds (0, 100, 200, 300, 400). In the revised paper, we have explicitly indicated the number of runs conducted for each experiment. Regarding the variance between different runs, we also added Table 2 and Table 3 to Appendix A.3 to thoroughly evaluate the effectiveness and robustness of PDT under different training configurations. The mean and standard deviation for all metrics are reported.
>
> We hope we have adequately addressed your concerns. Any further insights from you would be immensely valuable in improving our work.

---

### Official Review · Reviewer_hZ7t · 2024-11-04

**Soundness:** 2
**Presentation:** 4
**Contribution:** 3
**Rating:** 5
**Confidence:** 4

**Summary:**

This paper presents a new approach that treats deep neural network training as a nonlinear dynamical system, using Koopman operator theory to predict network weights several epochs ahead and bypass costly SGD iterations. To address issues like gradient explosion in large models, it introduces *predictive differential training* (PDT), where parts of the network learn at different rates. PDT includes a masking strategy to focus on parameters with reliable predictions and an acceleration scheduler to revert to traditional methods if needed. Compatible with optimizers like SGD and Adam, PDT demonstrates faster convergence, lower loss, and fewer epochs in experiments.

**Strengths:**

1. The paper's introduction is clear, and the effectiveness of the proposed algorithm is well illustrated in Figure 1. The integration of Koopman Operator Theory is smooth and accessible.

2. The understanding of the optimization method is accurate, and the motivating example in section 3.2 is clearly presented.

3. Table 1, showing runtime, is well-organized and effective for comparison.

**Weaknesses:**

1. The statement in section 3.3 may be inaccurate. $\mathcal{O}(N h^2)$ is indeed large, especially given the size of current deep learning models. Otherwise, methods like L-BFGS would be suitable for neural networks.

2. The experiments are limited to CIFAR-10, making it challenging to assess scalability to larger datasets. A drawback of these algorithms is that they often underperform compared to plain SGD on larger datasets.

3. What is the space complexity of the algorithm? Additionally, an ablation study on hyperparameters like batch size is missing, which is crucial for deep learning methods.

Minor Typo: Line 628 - there is "???"

**Questions:**

What motivates the use of the first principle? Is there any theoretical proof to support its validity? Traditional methods, such as line search, do not depend on this assumption.

Although the paper is well-written, the proposed algorithm is less convincing than expected (see previous comments). Specifically, two issues are concerning: (1) space complexity appears to be a significant problem, and (2) the experimental results are not sufficiently compelling.

---

> ### Author Response · Authors · 2024-11-24
> **Response to Reviewer hZ7t (Part1)**
>
> Thanks for your valuable review. We will address your concerns one by one in the following.
>
> **Weakness 1: The statement of computational complexity in Section 3.3**
>
> We appreciate the observation regarding the complexity analysis. We agree that for very large models with a large number of parameters $N$, the $O(Nh^2)$ complexity could become significant. However, in our experiments, we have intentionally set $h$ to a relatively small value (e.g., 5), which makes the quadratic impact of $h$ manageable compared to $N$. The amount of computation required to perform a prediction is comparable to that of a gradient descent operation. Furthermore, unlike L-BFGS, which needs to be performed at every iteration, PDT conducts DMD calculations much less frequently, only once every several epochs, rather than batch-level intervals, thus reducing the computational overhead significantly.
>
> For example, in our Cifar-10 experiments, when the batch size is 256, 20 gradient descent operations are required in each epoch; so for 50 epochs, a total of 1000 gradient descent operations are required. When starting from the 5th epoch, setting the prediction interval to 1, and $h$=5, only 23 DMD-based prediction operations are required in 50 epochs. In addition, as shown in Fig. 5 (figures in the first column), it takes a much smaller number of epochs for the proposed PDT to converge.
>
> When the model size is large (e.g., ViT), the computational overhead required by the baseline optimizer to perform a gradient descent operation will also be enormous. The advantage of the proposed PDT algorithm is that it can be used as a plug-in to achieve faster convergence and reduce the amount of computation required to reach convergence. It can also be seen from Table 1 that the runtime of each epoch of PDT is only slightly increased compared to the baseline.
>
> We have added Fig. 11 to Appendix A.4 to show a detailed analysis of the computational cost in terms of FLOPs. While the per-epoch computation of PDT is slightly higher (69.71 TFLOPs) than SGD (56.74 TFLOPs) due to the additional DMD calculations and prediction operations, it achieves faster convergence in terms of total computation, representing a 23.74% reduction in total computation cost compared to the baseline optimizer.
>
> **Weakness 2: Limitation of the dataset**
>
> We would like to point out that our experiments actually extend beyond CIFAR-10, including ImageNet-1K experiments with ResNet-50 (Fig 5(c) and Table 1), and ImageNet-1K experiments with ViT-Base (Fig 5(d) and Table 1). Both experiments show consistent improvements over baseline optimizers.
>
> **Weakness 3: Space complexity of the algorithm, and ablation study on hyperparameters**
>
> We thank the reviewer for highlighting these important aspects. The space complexity of our approach primarily arises from storing the past parameter snapshots for Koopman operator approximation, which scales as $O(Nh)$. We keep the h value small (5~10) to reduce computational costs. In addition, the memory usage is temporary and only during DMD computation. We mentioned in future work that using streaming DMD can reduce the memory footprint of constructing trajectory matrices and enhance computational efficiency.
>
> We have added additional experimental results in Appendix A.3 to thoroughly evaluate the effectiveness and robustness of PDT under different training configurations. Table 2 and Table 3 in Appendix A.3 show the impact of different batch sizes and learning rates on the performance of PDT. Table 4 shows the impact of different optimizers (SGD, SGD with Momentum, and Adam) on the performance of PDT.
>
> **Weakness 4: Typos**
>
> We have fixed typos in Algorithm 1.

---

> ### Author Response · Authors · 2024-11-24
> **Response to Reviewer hZ7t (Part2)**
>
> **Question 1: What motivates the use of the first principle**
>
> Our motivation for using Eqs. 8 and 9 is twofold: theoretical and empirical.
>
> Theoretical: This is rigidly supported by the Koopman operator theory. The first paragraph of Sec. 2 stated that Koopman analysis is the representation of a (possibly nonlinear) dynamical system as a linear operator on a typically infinite-dimensional space of functions (Mezic, 2021; 2005) and that it directly contrasts with standard linearization techniques that consider the dynamics only in a close neighborhood of some nominal solution. Indeed, Koopman analysis can yield linear operators that **accurately capture fundamentally nonlinear dynamics**. This is the theoretical underpinning of the proposed PDT. However, in reality, we cannot really work in an infinite dimensional space, which is why later in Sec. 2, we introduced dynamic mode decomposition (DMD) (Schmid, 2010) that identifies a suitable finite basis for representing the infinite-dimensional Koopman operator. It is in this step that potential prediction errors can be introduced, which leads to the proposed PDT where not all predicted weights are used as what the state-of-the-art prediction-based training does. Instead, we only *select* those weights with “good” quality predictions, i.e., when weight trajectories exhibit locally smooth dynamics, to accelerate the training process. That is, the masking strategy effectively compensates for the potential prediction errors DMD might introduce by using a finite-dimensional subspace to approximate the infinite-dimensional space.
>
> Empirical: Eq. 8 ensures that prediction only occurs when it can provide meaningful acceleration, otherwise it is better to use the baseline directly. Unlike line search which optimizes step size along a fixed direction, our criterion evaluates whether DMD prediction can provide faster progress than a single optimization step. The experimental results in Figs. 6 and 7 also demonstrate the rationality of our proposed mask strategy.
>
> **Question 2: Concerns about space complexity and experimental results**
>
> To demonstrate the effectiveness of proposed PDT, we implemented PDT across a variety of popular neural network architectures (Fully-Convolutional-Network (FCN), AlexNet, ResNet-50, and ViT-Base)  and datasets (CIFAR-10 and ImageNet-1K). We also used a range of optimizers, including SGD, SGD with momentum, and Adam, as shown in Fig. 5. We have added an “Effect of Training Hyperparameters” Section in Appendix A.3 to thoroughly evaluate the effectiveness and robustness of PDT under different batch sizes and learning rates. We hope we have addressed your concerns.

---

> > ### Comment · Reviewer_hZ7t · 2024-11-26
> > **Follow Up**
> >
> > Thank you for your response. I would like to quickly summarize the key points I wish to address:
> >
> > * As you may have noticed from my review, I personally appreciate this work. To be honest, if deep learning were not as popular as it is today, I believe methods like L-BFGS and other quasi-Newton methods might dominate the field.
> >
> > * I have no doubts about the computational complexity—the overhead is acceptable, and I fully agree with this. However, my concerns arise from two main aspects:
> >   * For large models, space complexity is always an issue. Even if $h = 5$, the complexity increases $25$ times, which is nearly unacceptable for deep learning tasks.
> >   * Additionally, stochasticity can often undermine the accuracy of quasi-Newton-based methods. This issue is particularly relevant in meta-learning, where a phenomenon known as short-horizon optimization arises. For more details, see [1].
> >
> > * Regarding the results from ImageNet-1K, I understand the setup. However, the reason I feel a larger-scale experiment is necessary is that, for ImageNet with small batch sizes, the i.i.d. assumption nearly breaks down. In such cases, will the algorithm still perform effectively? ImageNet-1K alone may not be sufficient to address this concern.
> >
> > Since the review period is extended we could still make discussion.
> >
> > [1] Wu, Yuhuai, et al. "Understanding short-horizon bias in stochastic meta-optimization." arXiv preprint arXiv:1803.02021 (2018).

---

> ### Author Response · Authors · 2024-11-28
>
> Thank you for sharing insights and summarizing concerns. We appreciate the additional feedback and the extra time given for continued discussion! In the following, we respond to the concerns pertaining to space complexity, stochasticity, and the necessity of large-scale experiments.
>
> **On space complexity:**
>
> The complexity analysis in Sec. 3.3 is for the computational complexity, $O(Nh^2)$, - not the space complexity. In standard DMD, the space complexity is $O(Nh)$ since the DMD algorithm has to store the previous $h$ epochs of weights. We agree with the reviewer that for large network models, even with a very small $h$ value, this would have been extremely storage-inefficient. Hence in Sec. 5, the Discussion section, we described one of the future works is to adopt the so-called streaming DMD (Hemati et al. 2014; Liew et al. 2022) that removes the direct correlation between history and the memory requirement, since it does not have to store historical weight values. We expect the usage of streaming DMD to largely save memory footprint without sacrificing any benefits that standard DMD brings to the proposed PDT.
>
> >Hemati, M. S., Williams, M. O., & Rowley, C. W. (2014). Dynamic mode decomposition for large and streaming datasets. Physics of Fluids, 26(11).
>
> > Liew, J., Göçmen, T., Lio, W. H., & Larsen, G. C. (2022). Streaming dynamic mode decomposition for short‐term forecasting in wind farms. Wind Energy, 25(4), 719-734.
>
> **On stochasticity:**
>
> We agree that stochastic effects, particularly in large-scale tasks or meta-learning, can impact the stability and effectiveness of the predictive methods. Our masking strategy explicitly addresses this by ensuring predictions are only applied where the predictive confidence is high (based on both quantity and direction criteria in Eq.8 and Eq.9). In addition, the rollback mechanism incorporated in our acceleration schedule is specifically designed to mitigate the cumulative effect of stochastic deviations. This ensures that even in scenarios with significant noise, the training process can revert to standard SGD to correct errors and prevent divergence.
>
> **On the necessity of large-scale experiments:**
>
> The reviewer raised a very valid point!  When the batch is too small for a diverse dataset like ImageNet, the weight updates could be chaotic since each consecutive batch is no longer an identical distribution. There are two experimental designs that can test this: 1) test PDT on a very large dataset like ImageNet-22K and 2) design a batching scheme to intentionally violate the i.i.d. assumption of each batch using a smaller dataset such as CIFAR-10. In the second design, we keep the normal batch size, but only put samples of the same class in the batch. We also randomize the batch sequence instead of using any fixed order so that there is no regular training set dynamics that DMD might pick up on. We chose the second design given the limited time but also because non-i.i.d. is guaranteed in the second design.
>
> We added Fig. 13 and Table 5 in Appendix A.6 that show the performance and runtime comparison between SGD and PDT under the non-i.i.d. setting using the second experimental design since non-i.i.d. is guaranteed. We preserve the original i.i.d. sampling of the test set. All experiments are repeated with five random seeds (0, 100, 200, 300, 400) to ensure statistical significance. From Fig. 13, we see that the variance of all the performance curves in the non-i.i.d. case is generally larger than those of the i.i.d. case. In addition, in the non-i.i.d. case, learning starts out much more slowly for both SGD and PDT and both take longer to converge. However, despite the challenging non-i.i.d. setup, PDT still achieves better performance than SGD in terms of faster convergence without sacrificing accuracy. This experiment further demonstrates that PDT’s advantage extends beyond standard i.i.d. training conditions, showing its robustness to challenging data sets where traditional assumptions about data distribution are violated.
>
> We are truly grateful for your feedback and hope we have addressed your concerns.

---

> > ### Comment · Reviewer_hZ7t · 2024-11-28
> > **Follow Up Again**
> >
> > Dear authors,
> >
> > I have read through your comments carefully and believe we share a common understanding of the work's weaknesses, strengths, and potential future directions. I don’t have any additional reviews, as the questions and answers provided are clear. At this point, determining whether the contribution of the work is strong enough might be best left to the meta-reviewers, who are more senior and should be able to gather the necessary insights from our discussion.
> >
> > Thank you,

---

> > > ### Author Response · Authors · 2024-11-30
> > >
> > > We have thoroughly enjoyed the discussion. Thanks for your time!

---

### Official Review · Reviewer_rXs5 · 2024-11-06

**Soundness:** 2
**Presentation:** 2
**Contribution:** 3
**Rating:** 6
**Confidence:** 4

**Summary:**

- This paper introduces a Koopman operator-based method, called Predictive Differential Training (PDT), which uses an adaptive mask to enhance and accelerate training dynamics. The authors empirically demonstrate that PDT can achieve performance improvements and faster convergence.

**Strengths:**

- This paper is well-written and presents a compelling approach to addressing the gradient explosion issue in Koopman-based methods.
- The authors empirically validate their method across various network architectures on CIFAR-10 and ImageNet, providing support for its effectiveness.

**Weaknesses:**

- The accuracy of recovering an approximated Koopman operator, as I understand, is influenced by (i) the noise amplitude (stochasticity) in the learning dynamics and (ii) the dataset's complexity. High stochasticity can obscure deterministic trends, and greater dataset complexity can impede adequate state space coverage within a few steps. These factors seem essential to comprehending the dynamics of PDT and determining the valid scope of the method. However, the paper does not fully address or justify these aspects

**Questions:**

### Major
- The accuracy of estimating the Koopman operator may depend on the batch size and learning rate, as these primarily influence the stochasticity of learning dynamics. I am curious whether the proposed method maintains effectiveness when using smaller batch sizes or higher learning rates

- It would be insightful to observe how the mask elements evolve during training. I expect that the number of active elements (i.e., those set to 1) might decrease over time, potentially making the "prediction" block redundant. Please clarify the scope and limitations of this method as they relate to this aspect.

- Please provide more detailed experimental information, including the hyper-parameters used in Figs. 2 and 5–8, batch size, the number of runs conducted to obtain the averaged results, and the criteria or process used for selecting these hyper-parameters.

- I am curious about the decision to end training before loss convergence in Fig. 5. For example, in Fig. 5(b), the test accuracy on CIFAR-10 appears substantially lower than widely reported values (≥ 95%). Further clarification on this would be helpful.

### Minor
- In Eq. (2), $x_0$ should be $x_i$.
- Please correct typos in Algorithm 1.

---

> ### Author Response · Authors · 2024-11-24
> **Response to Reviewer rXs5 (Part1)**
>
> Thanks for your valuable review. We will address your concerns one by one in the following.
>
> **Weakness: The influence of stochasticity in the learning dynamics and the dataset's complexity**
>
> We thank the reviewer for raising these important points about the theoretical foundations of our method. We agree that the effectiveness of Koopman-based prediction can be influenced by the stochasticity of learning dynamics and the dataset complexity. Although we did not explain our proposed PDT framework from this perspective, our algorithm actually circumvents this problem in a novel way, which also explains why the previous Koopman-based prediction method (i.e., prediction on all weights without the differential treatment) could not achieve good results on larger-scale networks. Our masking strategy inherently accounts for stochasticity by only selecting weights whose predicted trajectories show consistent behavior with the gradient descent baseline. The proposed PDT is integrated with the standard optimizer as a plug-in, allowing us to handle some unpredictable dynamics. The integration of standard optimization steps also provides a fallback mechanism when predictions become unreliable.
>
> We updated Fig. 2 by adding the performance curve of the proposed PDT, so the three curves represent SGD (baseline), prediction without selection, and PDT. It shows that as the depth of the network increases from two layers to six layers,  the prediction without selection performance curve starts showing extreme training loss, while PDT maintains stable performance.
>
> **Question 1: The effect of different batch sizes and learning rates**
>
> On the effect of batch size and learning rate, we have added a new section Appendix A.3, “Effect of Training Hyperparameters” to thoroughly evaluate the effectiveness and robustness of PDT under different training configurations. We include comprehensive experimental results across different learning rates from 0.001 to 0.1 (0.001, 0.01, 0.05, 0.1) and batch sizes from 32 to 512 (32, 64, 128, 256, 512). The results are shown in Tables 3 and 4 in Sec. A.3. All experiments were repeated with five random seeds (0, 100, 200, 300, 400) to ensure statistical significance.
>
> From Table 2, we see that at lower learning rates (0.001, 0.01, and 0.05), PDT consistently outperforms SGD in terms of convergence speed across different batch sizes. PDT shows a significant reduction in the runtime to reach baseline best loss, with an average runtime reduction of 22.76% compared to SGD. For higher learning rates (0.1), both SGD and PDT struggled to achieve stable training, and PDT's advantage over SGD became less pronounced, suggesting the high learning rate introduced significant stochasticity, reducing the effectiveness of PDT's prediction mechanism. Smaller batch sizes (32, 64) generally achieve more significant runtime reductions.
>
> To address the stability issues observed at higher learning rates and larger batch sizes, we further investigated the effectiveness of the learning rate scheduler and tested the Cosine Annealing learning rate scheduler with a minimum learning rate of 1e-3. The results are shown in Table 3. The results are particularly noteworthy at higher learning rates (lr=0.1), where the previous experiments in Table 2 show considerable variance. With the cosine annealing scheduler, PDT achieves consistent accuracy improvements across all learning rates while maintaining substantial runtime reductions, even with larger batch sizes.

---

> ### Author Response · Authors · 2024-11-24
> **Response to Reviewer rXs5 (Part2)**
>
> **Question 2: How do the mask elements evolve during training**
>
> Thank you for this insightful question about the evolution of mask elements. We have added the masked ratio vs. epoch curves to Fig. 5 as the last column. We made the following interesting observations, some of which can lead to exciting future works.
>
> - The masked ratio always starts with higher values in the early stage of the training process, then generally decreases as training progresses. This observation confirmed the reviewer’s hypothesis.
>
> - More interestingly, we observe that smaller networks on simpler tasks (FCN/AlexNet on CIFAR-10) show a relatively more gradual reduction in the masked ratio, while larger networks on more complex tasks (ResNet-50/ViT on ImageNet) exhibit a much sharper reduction of masked ratio, especially at the early stage of the training process. This pattern implies that for larger networks on larger datasets, the training dynamics are more complex and challenging to predict at the initial training stage, resulting in a rapid reduction of the percentage of weights that can be convincingly predicted (according to the proposed masking strategy). The training process of a deep network with millions to billions of parameters indeed presents an intriguing dynamical system that the control community has not faced before. This would stimulate further investigation into the development of better data-driven dynamical system analysis algorithms in addition to DMD.
>
> - More intriguingly, we observe that as training prolongs and as training loss converges to a stable value, we should expect the training dynamics to be less complex or easier to predict, which should have resulted in a higher masked ratio. However, in reality, except for the ResNet-50 on ImageNet-1K curve where a small bouncing back on the masked ratio is observed toward the end of the training process, all the rest of the scenarios exhibit a stable masked ratio, much lower as compared to that at the beginning of the training process. In addition, we would have expected the epoch number, where the masked ratio starts turning flat, to be consistent with that when the training loss enters a plateau, but this is only observed in the complex network scenarios [(c) and (d)], but not the simple network cases [(a) and (b)]. This seems to indicate that the masked ratio curves can have the potential of indicating when the network overfits, that when the marked ratio starts to drastically decrease again after the initial reduction. This would serve as a potential indicator for early stopping conditions. Although this is out of the scope of the current paper, the potential impact warrants further investigation. We thank the reviewer again for asking this insightful question.
>
> **Question 3: More detailed experimental information**
>
> We have included a more detailed description of the experimental configuration (dataset, network structure, batch size, learning rate, learning rate scheduler, etc.) in the caption of each figure and table. The default PDT-related hyperparameters used in most of the experiments (except for Fig. 9) were set to prediction step=5, prediction interval=1, start epoch=5, and past snapshot counts=5.
>
> Hyperparameters were initially determined through extensive experiments on CIFAR-10 with AlexNet, and then tested on other models. For general training hyperparameters, the learning rates were selected through grid search using different baseline optimizers. Batch sizes were chosen based on model size and GPU memory constraints: For CIFAR-10 experiments, batch sizes can be from 32 to 512 (32, 64, 128, 256, 512). For ImageNet experiments with ResNet-50 and ViT-base, we chose a batch size of 600 to ensure that there is enough GPU memory for DMD-based prediction calculations. We have added comprehensive experimental results in Appendix A.3 to thoroughly evaluate the effectiveness and robustness of PDT under different training configurations. Table 2 shows the impact of different batch sizes and learning rates on the performance of PDT. Table 3 explains why we used the cosine annealing learning rate scheduler to obtain more stable results.
>
> For PDT-related hyperparameters, we also conducted extensive experiments on Alexnet, as shown in Fig. 9. We finally settled on our default settings:
> - Prediction step=5: balances acceleration and stability. Larger values can lead to less accurate predictions, while smaller values limit acceleration benefits.
> - Past snapshot counts=5: balances computational costs and prediction accuracy.
> - Start epoch=5: allows the model to establish initial training dynamics before applying predictions.
> - Prediction interval=1: provides the best trade-off between computation cost and training acceleration.
>
> These parameter choices show good generalization performance across different models and tasks (as shown in Fig.5), as demonstrated by consistent performance improvements in our experimental results.

---

> ### Author Response · Authors · 2024-11-24
> **Response to Reviewer rXs5 (Part3)**
>
> **Question 4: Concern about the final test accuracy**
>
> In this paper, we focus our studies on the potential of the proposed PDT in accelerating training, i.e., using a smaller number of epochs to achieve the best training loss reported by baseline, without sacrificing performance (e.g., accuracy). We understand that this may have led to some confusion, especially regarding final accuracy values. The final accuracy values in this paper are slightly lower than reported values in other papers because we used basic architectures without modern augmentations and regularization techniques. For results in Fig. 5(b), we would like to point out that the achieved accuracy (around 82%) is consistent with most reported values (80-85%) for AlexNet on CIFAR-10 using SGD optimizer. There have indeed been accuracies reported as high as 99% on CIFAR-10, (<https://paperswithcode.com/sota/image-classification-on-cifar-10>), but they use a different network structure, or with a pretrained model. Please note that we develop PDT such that it can be plugged in any existing optimization framework.
>
> **Question 5: $x_0$ in Eq.(2)**
>
> In Eq. (2), $x_0$ uses the correct subscript.
>
> $g(x_{i+1}) = K g(x_i) = K K g(x_{i-1}) = … = K^{i+1} g(x_0) = \sum_{k=1}^\infty \lambda_k^{i+1} \phi_k(x_0) c_k$
>
> where $\lambda_k$ is an eigenvalue associated with the eigenfunction $\phi_k(\bf{x})$ evaluated at the initial condition $\phi_k(\bf{x_0})$. Please refer to (Mezic 2020) Eq. 6.
>
> > Mezić, I. (2020). Spectrum of the Koopman operator, spectral expansions in functional spaces, and state-space geometry. Journal of Nonlinear Science, 30(5), 2091-2145.
>
> **Question 6: Typos in Algorithm 1**
>
> We have fixed typos in Algorithm 1.

---

> > ### Comment · Reviewer_rXs5 · 2024-11-26
> >
> > I thank the authors for their thoughtful and detailed responses to my questions and concerns.
> >
> > Although my concerns regarding the final accuracy and the use of larger datasets remain, the authors have addressed many of my other concerns through additional experiments. Consequently, I am raising my score.

---

> > > ### Author Response · Authors · 2024-11-27
> > >
> > > Thank you for your thoughtful feedback and for taking the time to review our response. We sincerely appreciate the reviewer's comments, suggestions, and recognition of our rebuttal.

---

### Official Review · Reviewer_suTt · 2024-11-12

**Soundness:** 3
**Presentation:** 3
**Contribution:** 2
**Rating:** 6
**Confidence:** 3

**Summary:**

The paper proposes a Koopman operator based method to predict intermediate weight updates in SGD training runs. The authors present empirical evidence showing faster convergence with their proposed method. The main contributions compared to prior work are I) in adding a masking step that updates only parameters with good predictions from the Koopman operator and ii) a schedule to switch between Koopman based prediction and regular SGD updates.

**Strengths:**

The method descriptions are very clear and the paper is well written.

**Weaknesses:**

Two main concerns:

1. The main concern for the proposed method is the computational complexity of the method and the per FLOP gain. Since  $h$ has to be considerably small to minimize costs of approximating the Koopman operator, the per FLOP gain is not completely clear in the experimental results. Would the authors be able to provide Fig 5 with x-axis as the training FLOPs? What is the per FLOP generalization performance gain?

2. What is the difference between the proposed method and the DMD based method in [https://arxiv.org/pdf/2006.14371]? How would they compare?

Additional:
What is the variance between the Koopman and SGD training runs? E.g., Figure 5(d), the difference between AdamW and PDT looks considerably minimal, I wonder whether they are within the randomness between different SGD runs.

**Questions:**

From Fig 9, it is observable that PDT is sensitive to the scheduling hyperparameters. What is the variance in these runs? Also from Fig 9 c it seems that for most of the prediction starting points, the difference between SGD and PDT is negligible. Does that mean that the benefit of PDT is mostly in the starting epochs (around epoch 7 when the red line diverges)? This also indicates that the optimal pred starting epoch needs to be exhaustively searched to obtain benefits through PDT.

It might also be worth studying the training dynamics effect not only on the loss but properties of the learned function, e.g., sharpness/smoothness or grokking? [https://arxiv.org/pdf/2402.15555], [https://arxiv.org/pdf/2010.01412]

Can fig 2 be combined in one plot with solid and dashed line for SGD and PDT? Also what is the effect of the width of the network? How do different optimizers compare for the same network width-depth. How sensitive is it to the learning rates?

---

> ### Author Response · Authors · 2024-11-24
> **Response to Reviewer suTt (Part1)**
>
> Thanks for your valuable review. We will address your concerns one by one in the following.
>
> **Weakness 1: the computational complexity and the per FLOP gain**
>
> We thank the reviewer for this insightful comment about computational efficiency. While our paper reported runtime comparisons in Table 1, we agree that a FLOP-based analysis would provide additional clarity.
>
> The value of $h$ is kept small (5-10) for two reasons, 1) to reduce computational costs of DMD-based prediction, but more importantly, 2) to get a more precise estimate of the dynamics.
>
> The amount of computation required to perform a prediction is comparable to that of a gradient descent (GD) operation. It is important to note that the prediction based on DMD calculation is much less frequent (once for several epochs) compared to the standard gradient descent operations (multiple times per epoch, depending on the batch size). For example, in our Cifar-10 experiments, when the batch size is 256, 20 gradient descent operations are required in each epoch, so for 50 epochs, a total of 1000 gradient descent operations are required. When starting from the 5th epoch and setting the prediction interval to 1, only 23 DMD-based prediction operations are required in 50 epochs. In addition, it takes much less number of epochs for the proposed Predictive Differential Training (PDT) to converge. As shown in Table 1, based on “Time to Baseline Best Loss” and “Runtime per Epoch”, we could calculate the average number of epochs for the Baseline to converge is 2145.36/21.45 = 100, as compared to 1294.74/27.86 =  47 for PDT to converge.
>
> We have conducted a detailed analysis of the computational cost in terms of FLOPs and added Fig. 11 to Appendix A.4. Fig. 11 provides the performance comparison between SGD and the proposed PDT, with epochs and TFLOPs as x-axis, respectively. While the per-epoch computation of PDT is slightly higher (69.71 TFLOPs) than SGD (56.74 TFLOPs) due to the additional DMD calculations and prediction operations, it achieves faster convergence in terms of total computation. Specifically, PDT requires 2596.30 TFLOPs to reach the baseline's best loss, compared to SGD's 3404.32 TFLOPs, representing a 23.74% reduction in total computation cost. Moreover, PDT achieves better final accuracy (79.70% vs 78.75%) despite using fewer FLOPs to reach convergence. These results demonstrate that PDT achieves substantial per-FLOP efficiency gains despite the additional computation required for DMD-based operation.
>
> **Weakness 2: the difference between the proposed method and the other DMD based method**
>
> Unlike the approach in (Tano et al., 2020) which applies DMD predictions to all parameters, our proposed PDT introduces a masking strategy and selectively applies predictions only to those parameters that exhibit ‘good’ prediction performance, hence “differential” training. This selective approach mitigates the gradient explosion risk. This design is crucial for the scalability of Koopman-based methods to more complex architectures. To make the comparison more clear, we added the performance curve of the proposed PDT to Fig. 2, where the three curves represent SGD (baseline), prediction without selection (Tano et al., 2020), and PDT. It shows that as the depth of the network increases from two layers to six layers, (Tano et al., 2020) starts showing extreme training loss, while PDT maintains stable performance. This also explains why (Tano et al., 2020) mainly demonstrated results on a relatively small network (three hidden layers with 40, 200, and 1000 neurons, about 2.9 × 106 trainable parameters.), while we show generalization performance on modern architectures like ResNet-50 and ViT-base, as reported in Fig. 5.
>
> **Additional concern about variance of runs**
>
> All experimental results shown in Fig. 5 were repeated 5 times with different random seeds. Regarding the variance between different runs, we also added Table 2 to Appendix A.3, to thoroughly evaluate the effectiveness and robustness of PDT under different training configurations. All experiments were repeated with five random seeds (0, 100, 200, 300, 400) to ensure statistical significance. The mean and standard deviation for all metrics are reported in Table 2 and Table 3. The time to reach the baseline's best loss shows consistent acceleration. Taking the lr=0.05, batch size=256 configuration as an example, the difference in “Time to Baseline Best Loss” between SGD and PDT (648.39 seconds vs 507.62 seconds) is much larger than the standard deviations of both methods (8.57 seconds for SGD and 11.36 seconds for PDT), indicating that the acceleration effect is well beyond the statistical variance of different runs. PDT consistently outperforms SGD in terms of convergence speed across different batch sizes and learning rates, not merely due to randomness between different runs.

---

> ### Author Response · Authors · 2024-11-24
> **Response to Reviewer suTt (Part2)**
>
> **Question 1: Concern about prediction starting points**
>
> PDT is indeed affected by the value of the hyperparameters. However, as shown in Fig. 9, when the hyperparameters are chosen within a reasonable range, PDT can always maintain performance advantages. For example, Pred-steps =1~5 all show improved convergence over baseline.
>
> We ran each experiment 5 times with different random seeds, and the results were similar. We have added Table 2 in Appendix A.3 where mean and std are both shown.
>
> Regarding the prediction starting point shown in Fig. 9c, as described in Sec. 4.4, it needs to be greater than or equal to the number of epochs used to build the snapshot. In addition, since the training process usually displays higher degree of dynamics in the initial stage of training than later stage, the starting epoch needs to be as early as possible to be able to capture the dynamics more precisely. This observation is validated by the empirical results.
>
> **Question 2: The impact of the properties of the learned function**
>
> Thank you for your insightful suggestion. We agree that investigating the impact of PDT on the properties of the learned function, such as loss surface sharpness or smoothness, is highly valuable. These metrics provide a deeper understanding of the model's robustness and generalization capabilities. Based on the current experimental results, we think the selective application of predictions may help avoid sharp local minima by allowing more exploration in the weight space. In future work, we intend to incorporate these measures into our analysis to provide a more comprehensive evaluation of PDT, and further explore how these properties influence the efficacy of PDT. We have added this paragraph into Sec. 5 (Discussion).
>
> **Question 3:**
>
> The three subplots in Fig. 2 correspond to three network depths. We intend to use this figure to show why existing predictive training approaches like (Tano et al., 2020), i.e., prediction is applied to ALL network weights without selection, only works for smaller networks. The influence of network width is the same as the depth. It is the amount of weights that hinders these predictive training approaches from being effective on larger networks. See response to Weakness 2) above. We have updated Fig. 2 to include the performance of PDT on fully connected networks with different depths, to better demonstrate the advantages of PDT.
>
> We have added additional experimental results in Appendix A.3 to thoroughly evaluate the effectiveness and robustness of PDT under different training configurations. Table 2 and Table 3 in Appendix A.3 show the impact of different batch sizes and learning rates on the performance of PDT. Table 4 shows the impact of different optimizers (SGD, SGD with Momentum, and Adam) on the performance of PDT, trained on the same network structure with the same width and depth. All these results show consistent performance gain of PDT over existing optimizers.

---

> > ### Comment · Reviewer_suTt · 2024-11-29
> > **Thanks for the response**
> >
> > I thank the authors for their response, they have provided experimental evidence that clears my concerns. Therefore I'm increasing my score.

---

> > > ### Author Response · Authors · 2024-11-30
> > >
> > > Thank you for taking the time to review our response. We sincerely appreciate your comments, suggestions, and recognition of our rebuttal.

---

### Author Response · Authors · 2024-12-04
**Summary of Rebuttal and Revisions**

We would like to thank all the reviewers for your time, the weaknesses pointed out, and the insightful questions. All these have greatly helped us improve the quality of the work. Here, we summarize the changes made to the paper - results from this 3-week discussion phase that we thoroughly enjoyed:

1. Added comprehensive experimental results across different batch sizes, learning rates, and optimizers. (Appendix A.3)
2. Added Detailed computational complexity and FLOP analysis. (Appendix A.4)
3. Added non-i.i.d. training experiments to test robustness. (Appendix A.6)
4. Modified figures and tables to show not only the mean values but also standard deviations from multiple runs.
5. Added a more detailed analysis of the masked ratio in experiments. Clarified how PDT differs from prior work by preventing gradient explosion through selective prediction, enabling scalability to larger models, and handling stochasticity.
6. Added a discussion on leveraging streaming DMD to further reduce space complexity and scale to even larger datasets and architectures.

We hope these revisions and clarifications during rebuttal sufficiently address the reviewers' concerns. We believe the extensive additional analysis and the consistent performance gains observed across different architectures, optimizers, and training conditions demonstrate PDT's potential as a broadly applicable training acceleration technique.

---

### Meta-Review · Area_Chair_VQea · 2024-12-16

**Metareview:**

The paper proposes a new learning strategy based on Koopman operators. Unlike previous methods that struggled with gradient explosion issues, the proposed framework is more stable and can be applied efficiently to more complex tasks and architectures. The approach introduces Predictive Differential Training, which uses a masking strategy to enhance stability and improve training efficiency. Additionally, it can be integrated into existing optimizers such as SGD and Adam. Experimental results demonstrate that the method accelerates convergence and reduces training time, although concerns remain regarding its generalizability to larger datasets and architectures.

**Additional Comments On Reviewer Discussion:**

The paper introduces an interesting concept with a solid theoretical foundation, but key empirical concerns remain unresolved, which are critical for recommending acceptance given the scope of the work.

While the authors responded to several reviewer concerns during the discussion period, the lack of comprehensive experimentation on larger datasets and architectures, as well as lingering doubts about the algorithm’s general behavior, prevent the paper from meeting the necessary standards for acceptance. In particular:

* Reviewers highlighted the need for testing on **larger datasets** beyond CIFAR-10 and ImageNet-1K. The current experiments show inconsistent gains, and even with additional results, the question of generalizability to larger architectures and datasets remains insufficiently addressed. Scalability tests on larger benchmarks like ImageNet-21K or other complex datasets are crucial for demonstrating the method's practical utility.
* There are still concerns about **overfitting**, as indicated by the test loss plots (Fig. 5). Additionally, the decision to stop training before loss convergence raises doubts about the method’s effectiveness. A clearer explanation of this choice, along with adjustments to the training process, would help solidify the paper’s claims.

I encourage the authors to continue refining the empirical validation, address the scalability concerns, and better integrate the appendix material into the main text in future submissions.

---

### Decision · Program_Chairs · 2025-01-22

Reject